# In Silico and In Vitro Analyses of Strawberry-Derived Extracts in Relation to Key Compounds’ Metabolic and Anti-Tumor Effects

**DOI:** 10.3390/ijms26083492

**Published:** 2025-04-08

**Authors:** Lucia Camelia Pirvu, Amalia Stefaniu, Sultana Nita, Nicoleta Radu, Georgeta Neagu

**Affiliations:** 1Department of Pharmaceutical Biotechnologies, National Institute for Chemical Pharmaceutical Research and Development (INCDCF-ICCF), 112 Vitan, 031299 Bucharest, Romania; astefaniu@gmail.com; 2Department of Physical-Chemical Analysis and Quality Control, National Institute for Chemical Pharmaceutical Research and Development (INCDCF-ICCF), 112 Vitan, 031299 Bucharest, Romania; sultananita@ncpri.ro; 3Biotechnology Faculty, University of Agronomic Sciences and Veterinary Medicine of Bucharest, 59 Marasti, District 1, 011464 Bucharest, Romania; nicoleta.radu@biotehnologii.usamv.ro; 4Department of Biotechnology, National Institute of Chemistry and Petrochemistry Research and Development, 202 Splaiul Independentei, 060021 Bucharest, Romania; 5Department of Pharmacology, National Institute for Chemical Pharmaceutical Research and Development (INCDCF-ICCF), 112 Vitan, 031299 Bucharest, Romania

**Keywords:** 40% ethanolic extract from strawberry fruits, phloridzin phloretin and 4-methylchalcone, 2,3-dihydro-3,5-dihydroxy-6-methyl-4H-pyran-4-one 2-pyrrolidinone 5-(cyclohexylmethyl) and hexadecanoic acid, in silico SwissADME analysis, in silico CLC docking analysis for Bcl-2 TNKS and COX-2 targets, in vitro cytotoxicity and anti-proliferative studies, Caco-2

## Abstract

Plant extracts contain many small molecules that are less investigated. The present paper aims to study in silico physical-chemical, pharmacokinetic, medicinal chemistry and lead/drug-likeness properties and the ability to interfere with the activity of P-glycoprotein (P-gp) transporter and cytochrome P450 (CYP) oxidase system in humans of phloridzin, phloretin, 4-methylchalcone metabolic series alongside the top three compounds found in the ethanolic extract from strawberries (S), namely 2,3-dihydro-3,5-dihydroxy-6-methyl-4H-pyran-4-one, 2-pyrrolidinone 5-(cyclohexylmethyl) and hexadecanoic acid. The phloridzin derivatives also were studied for their inhibitory potential upon Bcl-2, TNKS1 and COX-2 molecular targets. In vitro, Caco-2 studies analyzed the cytoprotective and anti-proliferative activity of S and the three phloridzin derivatives (pure compounds) in comparison with their combination 1:1 (GAE/pure compound, *w*/*w*), in the range 1 to 50 µg active compounds per test sample. Altogether, it was concluded that phloretin (Phl) can be used alone or in combination with S to support intestinal cell health in humans. Phloridzin (Phd) and phloridzin combined with S were proven ineffective. 4-methylchalcone (4-MeCh) combined with S indicated no advantages, while the pure compound exhibited augmented inhibitory effects, becoming a candidate for combinations with anticancer drugs. Overall, in silico studies revealed possible limitations in the practical use of phloridzin derivatives due to their potential to interfere with the activity of several major CYP enzymes.

## 1. Introduction

Chalcone derivatives are among the most active compounds of the flavonoid subclasses [1,2]. However, the screening studies intended to quantify the effects of chalcone derivatives within a plant extract are scarce; one possible approach is to design experiments using controlled amounts of chalcone derivatives added to a specific plant extract. This strategy helps to better elucidate and quantify their potential cytoprotective or cytotoxic effects. Furthermore, chalcone derivatives are less stable and are typically present in very small quantities in green plants, compared to other flavonoid subclasses; hence, they are likely to be lost during the technological processes. As a possible solution, adding pure chalcones to a final vegetal extract (plant-derived product) could lead to a more stable combination than the initial product.

Besides, vegetal extracts contain many other small understudied active molecules. For example, the gas-chromatographic (GC-MS) analysis of the 40% ethanolic extract from strawberry fruits in the present study has revealed three main small volatilizable compounds of real pharmacological interest: 2,3-dihydro-3,5-dihydroxy-6- methyl-4H- pyran-4-one, 2-pyrrolidinone 5-(cyclohexylmethyl) and hexadecanoic acid, respectively.

Among these, the compound 2,3-dihydro-3,5-dihydroxy-6-methyl-4H-pyran-4-one has also been reported as the primary compound in the ethanolic extract from *Mylochia pyramidata* L. [3]. *Calendula officinalis* L. extracts [4] as well as the culture supernatants from *Lactobacillus pentosus* S-PT84 [5] also indicated the presence of this small compound. Pharmacologically, it was assigned with certain antimicrobial activity upon the *Pseudomonas aeruginosa*, *Staphylococcus aureus* and *Candida albicans* strains [3,4], with stimulatory activity upon the autonomic nervous system in rats [5], as well as with promising anti-inflammatory and antioxidant effects being proposed for further applications in alleviating osteoporosis, diabetes and cardiovascular maladies in humans [6]. The compound 2-pyrrolidinone 5-(cyclohexylmethyl), identified by GC-MS analysis in the chloroform fraction from the leaves of *Rubus steudneri*, was proved with significant cytotoxic effects and cell cycle arrest activity on the breast cancer cell line Michigan Cancer Foundation-7 (MCF-7) [7]. N-hexadecanoic acid isolated from the chloroform extract of *Kigelia pinnata* dry leaf powder exhibited a high affinity for DNA topoisomerase-I and high activity against the human colorectal carcinoma cell line HCT-116 (IC_50_ = 0.8 µg/mL) [8]. Considering their common presence in vegetal extracts, it results that these small compounds could significantly contribute to the pharmacological activity of so-called polyphenolic concentrates from medicinal plants and vegetal food.

Regarding strawberry fruit-derived products, besides their well-known nutritional qualities [9,10,11,12,13,14], these also contain numerous compounds in the series of secondary metabolites with well-established pharmacological benefits. Proving these, analytical studies on 27 *Fragaria ananassa* cultivars revealed up to 133 mg total phenolics (gallic acid equivalents, GAE Eq) per 100 g fresh fruits (FW) [12]; pelargonidin-3-*O*-glucoside, pelargonidin-3-*O*-rutinoside, cyanidin-3-*O*-glucoside, catechin and epicatechin are the main phenolic compounds in strawberry fruits, achieving up to 73 mg per kg (FW) [12,13]. The variety *F. ananassa* Alba can reach up to 226 mg GAE Eq per 100 kg FW [13]. Smaller quantities of quercetin-3-*O*-glucopyranoside, quercetin-3-*O*-rhamnopyranoside, quercetin and kaempferol-3-*O*-glucopyranuronides, quercetin and kaempferol aglycones, aside 3,4,5-methoxycinnamic, *p*-coumaric, ellagic, protocatechuic, salicylic, vanillic, caffeic, chlorogenic, cinnamic, ferulic and isoferulic acids, plus pyrogallol, catechol and coumarin also were proven in strawberry fruit-derived products [15,16]. Besides, strawberries are a natural source of dihydrochalcone derivatives, mainly of phloridzin glycoside. Phloridzin is the chemotaxonomic marker for the *Fragaria* genus, being used to establish the authenticity of the strawberry-derived products; phloridzin content in strawberries is estimated at 1.9 to 4.9 mg per 100 g dry weight (DW) [17,18].

Altogether, the secondary metabolites found in strawberries and strawberry-derived products are in the series of highly active natural vegetal compounds. Chalcone derivatives, particularly their aglycones are proven with a plethora of valuable pharmacological activities, which recommends them for many practical applications. Among these, phloridzin, the dominant chalcone derivative found in green plants, has limited practical utilization due to its very low bioavailability in humans (8.67%) [19,20]. At the opposite pole, the aglycone of phloridzin, phloretin is proven with high bioavailability in humans and holistic anti-inflammatory, antimicrobial, anti-tumor, anti-proliferative, angiogenic and anti-metastatic properties, at the same time being able to interfere with the glucose transport in cells [21,22,23,24]; summing together, phloretin checks most of the features requested for a feasible anti-tumor candidate. Similarly, the methylated chalcone derivatives also appear as promising drug candidates [25,26].

Proving these, starting from a natural chalcone derivative isolated from *Glycyrrhiza* species, studies on a collection of newly developed α-methylchalcone derivatives [27] indicated the ability of 3,4-O-methyl derivative in position R4 (encoded 3k) to induce augmented inhibitory effects on the cervical cancer cell lines HeLa and HeLa/DDP, at the same time low toxic effects on the normal line H8. The anti-tumor activity has been developed through stimulatory effects upon the cell apoptosis concomitantly with cell growth inhibition in the G2/M phase, together decreasing the invasion and the migration of the cancer cells. Furthermore, by combining cisplatin with 3k, a significant reduction in cell resistance to chemotherapy has also been noticed; the resistance index decreased from 7.90 to 2.10. It was also observed that 3k did not affect the expression of the protein transporter (P-gp) in cells and effectively decreased the fluorescence intensity of the α and β microtubules in HeLa and HeLa/DDP cells, finally inducing the disruption of the cell morphology, the reduction of the living cells and the coagulation of the nucleus, too. The Western blot analysis confirmed the ability of the methylated derivative to significantly reduce the levels of the polymerized microtubule proteins, in both HeLa and HeLa/DDP cell lines, while concurrently increasing the expression of the dissociated α and β microtubule proteins. Finally, in silico studies (CADD docking analysis) anticipated the ability of the methylated derivative to perform stable hydrogen bonding with the microtubule proteins in cells [27], thus sustaining the usefulness of the computational studies in expanding the knowledge in the field of natural compounds.

The potential to induce synergistic pharmacological effects has also been proven for chalcone derivatives. Studies [28] on the Lewis lung cancer (LLC) xenograft model revealed that the Lewis cells exposed to phloretin from 25 to 200 μg pure compound inhibited cell proliferation in a time- and dose-dependent manner; also, it was proved that combining phloretin and radiotherapy the survival of the cells and the apoptosis significantly increased, while the proliferation index decreased, by comparison with both, the negative control series and the positive (phloretin) control series, too. The synergism between phloretin and radiotherapy has been developed by cumulative gains in apoptosis and cell viability and by reducing the glucose transport in cells. Similarly, studies aimed to test the efficacy of several natural food additives (for maintaining the quality of the fresh noodles) indicated that combining myricetin with phloretin results in synergistic effects against the *Escherichia coli* and *Staphylococcus aureus* strains [29]; the synergistic effects were situated from one to ten times compared to the negative control samples, while the antimicrobial mechanism was attributed to the ability of the myricetin–phloretin combination to inhibit the activity of the ATPase.

Furthermore, starting from the observation of high antioxidant and ROS efficacy of the strawberry-derived products (proved active on peroxyl (ROO*), superoxide (O^2−^), hydrogen peroxide (H_2_O_2_), hydroxyl (*OH) and singlet oxygen (^1^O_2_) radicals [30,31,32,33]), some in silico studies proved the potentiality of arbutin derivatives (mainly found in strawberry tree) and other key phenolics from strawberries to impact the activity of the tyrosinase enzyme (TYR) [34]. This is in the context in which TYR is involved in the generation of dopamine-quinone derivatives assigned a major role in oxidative stress associated with different health disorders in humans. Closely connected with the antioxidant activity, the cardiovascular benefits of the strawberry-derived products also tried to be explained by in silico tools; thus, the major phenolic derivatives, ellagitannin and pelargonidin-3-glucoside were tested for their ability to impact the activity of the angiotensin-converting enzyme (ACE) [35]. In the specific case of dihydrochalcone derivatives, given their ability to manage the glucose level in cells, in silico studies were focused on the main protein disease and molecular targets involved in diabetes: e.g., α-glucosidase, Aldose Reductase (ALR), Glucose Transporter Type 4 (GLUT4), Sodium Glucose Cotransporter 2 (SGLT2), Dipeptidyl Peptidase 4 (DPP-4), Protein Tyrosine Phosphatase 1B (PTP1B), Peroxisome Proliferator-activated Receptor-gamma (PPARγ) and Adenosine Monophosphate (AMP)-activated Protein Kinase (AMPK) [36]. The current study designed in silico investigations upon the phloridzin metabolic series to estimate their potentiality to impact three molecular targets concerned with tumor initiation and tumor progress in humans: B-cell lymphoma protein (Bcl-2), human tankyrase 1 (TNKS1) and cyclooxygenase 2 (COX-2), respectively.

Altogether, the current studies aimed to carry out in silico investigations on three main chalcone derivatives (phloridzin, phloretin and 4-methylchalcone) alongside three main small compounds identified in strawberry fruit ethanolic extract (2,3-dihydro-3,5-dihydroxy-6-methyl-4H-pyran-4-one, 2-pyrrolidinon 5-(cyclohexylmethyl) and hexadecanoic acid) to compare their physical-chemical, pharmacokinetic, medicinal chemistry and lead-likeness parameters as well as their ability to interfere with the activity of P-glycoprotein (P-gp) transporter and the cytochrome P450 oxidase system in humans. In the specific case of the three chalcone derivatives, it was also analyzed their ability to interact with three disease proteins associated with tumorigenesis and intestinal malignancy in humans, Bcl-2, TNKS1 and COX-2, respectively. Secondly, in vitro studies aimed to investigate the cytotoxic and the anti-proliferative effects of 40% ethanolic extract from strawberries, of the three chalcone derivatives (pure compounds prepared in 50% ethanol) as well as of their combinations made in a manner to ensure a 1:1 (*w*/*w*) ratio between the total phenolics in strawberry extract (expressed as GAE Eq) and each one chalcone derivative (pure compounds in the series of reference substance/r.s. for analytical purposes) upon the viability of Caco-2 cells. The results were compared with the negative control series consisting of cells exposed to the Caco-2 medium and cells exposed to the solvent in which the strawberry extract and the three pure compounds were prepared, respectively.

Being in the category of xenobiotics, the secondary metabolites from plants will interact with the cytochrome P450 (CYP) oxidase system in humans; at the same time, they could impact the activity of the CYP enzymes [37,38]. Such interactions are very difficult to investigate in vitro, and even less so in vivo, while in silico studies can provide important insights regarding their potential limitations in human use.

## 2. Results

### 2.1. Analytical Results

Table 1 presents the gas chromatography–mass spectrometry (GC-MS) results on the 40% ethanolic extract from strawberry fruits. The GC-MS analysis was driven by CLARUS 500 (PerkinElmerR, Waltham, MA, USA) apparatus as described in Section 4.3.

### 2.2. In Silico SwissADME Computational Results

In silico SwissADME computational studies [39] aimed to analyze physical-chemical, pharmacokinetic, medicinal chemistry and lead/drug-likeness parameters as well as the ability to interfere with the activity of P-glycoprotein (P-gp) transporter and the cytochrome P450 oxidase system in humans of the six compounds representative for strawberry derived products: phloretin-3-*O*-glucoside also known as phloridzin (Phd), the aglycone of phloridzin namely phloretin (Phl) and the metabolite of these two compounds in humans namely 4-methylchalcone (4-MeCh), alongside the three small compounds identified by GC-MS analyses as top compounds in 40% ethanolic extract from strawberry fruits, namely 2,3-dihydro-3,5-dihydroxy-6-methyl-4H-pyran-4-one, 2-pyrrolidinon 5-(cyclohexylmethyl) and hexadecanoic acid, respectively.

Figure 1 presents the chemical structures [39] of the phloridzin derivatives (a) and the three top compounds identified in 40% ethanolic extract from strawberries (b).

Table 2 presents in silico SwissADME comparative analysis [40,41,42,43,44] on the three naturally occurring phloridzin derivatives (Table 2a) and the three top compounds identified in 40% ethanolic extract from strawberries (Table 2b).

On the basis of cumulative computational results, particularly by comparing the topological polar surface area (TPSA), lipophilicity and hydrophilicity properties of the three chalcone derivatives (Table 2a), it resulted that phloridzin might offer a wider palette of active formulas since “having a soluble molecule greatly facilitates many drug development activities, primarily ease of handling and formulation” [42].

In the case of the three small compounds from the strawberry ethanolic extract (Table 2b), 2,3-dihydro-3,5-dihydroxy-6-methyl-4H-pyran-4-one appears to have better formulation features, also attributable to its higher solubility in water [45].

Regarding their most probable fate during the digestion process in humans [46], the BOILED-Egg representation [41], Figure 2 depicts the likelihood of passive absorption through the gastrointestinal (GI) mucosa and the blood–brain barrier (BBB). As stated, the compounds located outside the egg are unlikely to pass through the GI mucosa; the compounds in the white region are likely to be passively absorbed, while the compounds in the yellow region have a high probability of crossing the BBB.

The BOILED-Egg representation also offers information about the ability of small compounds to act as P-glycoprotein (P-gp) transporter substrate in humans: the PGP+ are labeled as blue dots and the PGP− are labeled as red dots. Summing together, phloridzin (1) indicates a low probability of crossing the GI barrier in humans, and no expectation to pass the BBB. Phloretin (2) and 2,3-dihydro-3,5-dihydroxy-6-methyl- 4H-pyran-4-one (4) reveal a high probability of passing through the GI barrier, but not the BBB. 4-Methylchalcone derivative (3) similar to 2-pyrrolidinone 5-(cyclohexylmethyl) (5) and hexadecanoic acid (6), each one indicates the ability to cross the BBB, and also the GI mucosa in humans. Finally, phloridzin reveals the ability to act as a P-gp substrate, while the other five test compounds do not interact with the P-gp transporter in humans. It must be noted that using another web tool [47] and interpretation data [48,49,50,51], it resulted that phloretin (logP = 2.05) can cross the blood–brain barrier in humans [52].

Furthermore, plant compounds are xenobiotics and they will invariably interact with the cytochrome P450 (CYP) oxidase system in humans. As is well known, CYP enzymes are membrane-bound proteins found in all cells; these are especially important in the Phase I metabolism process of drugs and xenobiotics in humans [53,54,55,56,57,58,59,60,61,62,63,64,65,66,67,68,69,70,71,72,73,74,75,76]. Based on a tremendous number of validated experiments, the SwissADME web tool offers information on whether a given compound might inhibit five primary CYP isoforms in humans: CYP1A2, CYP2C9, CYP3A4, CYP2C19 and CYP2D6, respectively [53,54].

This way, the computation on the six small compounds under study indicated that phloretin can interfere with the activity of three Phase I CYP enzymes: CYP1A2 [55,56,57,58,59], CYP2C9 [60,61,62,63] and CYP3A4 [64,65,66,67,68,69], respectively. The 4-methylchalcone derivative also indicated the potentiality to inhibit CYP2C19 [70,71,72] and CYP2D6 [73,74,75,76] isoforms, while the hexadecanoic acid showed the ability to inhibit the activity of CYP1A2 [55,56,57,58,59] and CYP2C9 [60,61,62,63] isoforms. 2,3-dihydro-3,5-dihydroxy-6-methyl-4H-pyran-4-one and 2-pyrrolidinone 5-(cyclohexylmethyl) indicated the lack of interferences, both with CYP enzymes and P-gp transporter. Phloridzin also does not inhibit the activity of any of the five CYP isoforms, but has shown the ability to interfere with the activity of the P-gp transporter. As stated, a P-gp substrate is “a substance that uses the P-glycoprotein transporter for various activities, including drug absorption and drug excretion, which can lead to changes in the body or to some induced effects of other drugs on the body” [77]. This fact is especially important for phloridzin glycoside since it largely remains available along the GI tract.

From the medicinal chemistry standpoint, the computation studies yielded zero PAINS alerts for all tested compounds; as stated, PAINS are compounds that interact non-specifically with multiple molecular targets causing false-positive results in computational screens [78,79]. Finally, one Brenk, Michael acceptor alert was noted for 4-methylchalcone [41,80,81] derivative; Michael acceptor alert refers to compounds that contain an α,β-unsaturated carbonyl fragment able to act as an electrophilic acceptor in Michael 1,4 addition reactions. These can bind nucleophilic residues in a protein active site potentially offering therapeutic benefits, particularly in an anticancer context [80,81].

Regarding a compound’s likelihood of good oral bioavailability in humans [82,83,84,85], the SwissADME web tool allows the evaluation of both, “lead-likeness” (similarity to small molecule ligands) and “drug-likeness” (resemblance to known drugs) of the small compounds. The drug-likeness is computed on the basis of five notorious filters, specifically Lipinski, Ghose, Veber, Egan and Muegge [86,87,88,89,90,91,92].

Among the six compounds tested, phloretin only has revealed similarity to other molecule ligands, and passed all the five filters utilized. The other five compounds did not reveal similarity with known leads, but passed two or more test filters. The 4-methylchalcone derivative passed Lipinski, Ghose, Veber and Egan filters, 2-pyrrolidinone 5-(cyclohexylmethyl) passed Lipinski, Ghose, Veber and Egan filters, 2,3-dihydro-3,5-dihydroxy-6-methyl-4H-pyran-4-one passed Lipinski, Veber and Egan filters, the hexadecanoic acid passed Lipinski, Ghose and Veber filters, while phloridzin passed Lipinski and Ghose filters, respectively.

The punctual physical-chemical parameters are as follows: the calculated logP less than 5, the molecular weight less than 500, no more than 5 H-bond donors and no more than 10 H-bond acceptors in the case of Lipinski’s filter (also known as the rule of five) [86,87]; the molecular weight between 160 and 480, molar refractivity between 40 and 130, predicted logP between −0.6 and 5.6, and a total number of atoms between 20 and 70 in the case of Ghose’s filter [88]; less than 10 rotatable bonds and a polar surface area less than 140A2 in the case of Veber’s filter [89]; the molecular weight from 200 to 600 Daltons, a logP value from −0.6 to 5.6, a polar surface area (tPSA) less than 132A2, a predicted logP between −0.4 and 5.0, less than or equal to six H-bond donors, less than or equal to five H-bond acceptors in the case of Egan’s filter [90]; the molecular weight from 200 to 600 Daltons, a logP value from −2 to 5.0, a polar surface area (tPSA) less than 150A2, a predicted logP between −0.4 and 5.0, a number of rings less or equal with 7, a number of carbons (NC) greater than 4, a number of heteroatoms (NH) > 1 and RB ≤ 15, HBD ≤ 5 and HBA ≤ 10 cumulates the criteria of the Muegge’ filter [91,92].

Based on the cumulative criteria, these filters estimate small compounds’ profiles for further drug development. The SwissADME bioavailability radar (Figure 3) combines six physical-chemical parameters (lipophilicity, size, polarity, solubility, saturation and flexibility) to predict whether a compound might be a suitable lead [42].

As can be seen in Figure 3, none of the three chalcone derivatives is situated in the optimal area of the SwissADME radar: phloretin aglycone is too polar, while phloridzin and 4-methylchalcone are too unsaturated. Of the three smaller compounds from strawberry fruit, the hexadecanoic acid is too lipophilic and also too flexible; 2,3-dihydro-3,5-dihydroxy-6-methyl-4H-pyran-4-one and 2-pyrrolidinone 5-(cyclohexyl methyl) fits within the pink area, but their low molecular weight (MW < 200) disqualifies them as lead candidates. However, chemical modifications could address this limitation.

### 2.3. In Silico Molecular CLC Docking Results

Molecular docking studies were carried out by CLC Drug Discovery Workbench 2.4 software (QIAGEN, Aarhus, Denmark) [47], as described in the former studies [52,93,94]. Docking simulations aimed to investigate the potentiality of the three chalcone derivatives to interact with three protein–disease and molecular targets concerning intestinal malignancy in humans: Bcl-2, TNKS1 and COX-2. Studies were carried out in comparison with the native ligands, meaning validated inhibitors retrieved from the Protein Data Bank [95]. The three validated ligands (native inhibitors) were as follows: the PDB ID: 2O2F corresponding to the Bcl-2 target in complex to the native inhibitor namely the co-crystallized ligand LI0 [96]; the PDB ID: 4W6E corresponding to the TNKS1 target in complex to the native inhibitor namely co-crystallized 3J5A, a quinazoline derivative [97]; the PDB ID: 1PXX corresponding to the COX-2 target in complex to the diclofenac [98]. The ligands’ preparation was achieved by energy minimization using the Spartan’14 software program from Wavefunction, Inc., Irvine, CA, USA [99].

Table 3, Table 4 and Table 5 show the results of docking simulations on the three molecular targets (Bcl-2, TNKS1, COX-2) against their native ligands (LI0, 3J5A and diclofenac) in comparison with the three chalcone derivatives under study. The labeling of the chalcone derivatives ‘atoms is arbitrarily chosen upon structure generation by Spartan’ 14 mechanistic software. Their minimized structures including atom labels, are given in Appendix A.

From Table 3, the docking results on the Bcl-2 protein–disease target, it can be seen that the native ligand (the co-crystallized LI0 A 1000) exhibits the greatest inhibitory score (−82.97), achieving a stable complex by establishing one single hydrogen bond with TRP141 amino acid residue in the active pocket, on chain A. By comparison, phloridzin establishes six hydrogen bonds within the active site involving PHE109, ASP108, ARG143, LY142 and VAL130 amino acids residues on chain A and docking score of −62.85/0.09; phloretin establishes four hydrogen bonds with PHE109, VAL130 and ASP10 amino acids residues and reveals a lower docking score (−51.97/0.15). On the contrary, 4-methylchalcone is not able to hydrogen bonding, but another type of interaction within the interacting amino acids on chain A is quite sufficient to achieve a similar docking score (−52.09/0.10) and the quality of milder inhibitor of BCL-2 tumor target.

Docking results on the TNKS1 (Table 4) indicated the native ligand (the co-crystallized 3J5A) with the greatest inhibitory score (−104.15); this high score is based on 4 hydrogen bonds with 3 amino acid residues in the active pocket: (GLU1291, SER1221 and GLY1185). The three chalcone derivatives also revealed the potentiality to inhibit the TNKS1 activity, but more mildly: phloridzin indicated the higher inhibitory score (−74.58), followed by 4-methylchalcone (−67.80) and phloretin (−67.28).

Docking results on the COX-2 (Table 5) indicated the co-crystallized 3J5A having the greatest inhibitory score (−68.76); this high score is based on 3 hydrogen bonds with 2 amino acid residues (SER530, TYR385) in the active pocket. The other ligand chalcone derivatives also revealed the potentiality to inhibit the cyclooxygenase-1′ activity, revealing similar docking scores (−52.56 for phloridzin and −57.24 for phloretin, respectively). Although 4-methylchalcone is not capable of forming hydrogen bonding, its docking score (−58.38) is good enough to provide an inhibitory activity against COX-2.

Altogether, the docking scores suggested the ability of all three chalcone derivatives to inhibit all three molecular targets involved in tumorigenesis in humans.

### 2.4. Pharmacological Results

#### In Vitro MTS Cytotoxicity and Anti-Proliferative Assessments

In vitro studies were performed on the Caco-2 cell line (ATCC, HTB-37) following the Promega protocols for cytotoxicity and anti-proliferative MTS assays [100], respectively.

There were investigated the effects of 40% ethanolic extract from strawberries (S), of phloridzin (Phd), of phloretin (Phl) and of 4-methylchalcone (4-MeCh) pure compounds prepared in 50% ethanol, as well as of their combination (S-Phd, S-Phl, S-4MeCh) made in a manner to obtain the rate 1:1 (*w*/*w*) between the total phenolics in strawberry extract and the chalcone derivative in pure compounds’ solutions, at the same time to assure the concentration scale from 1 to 50 µg total active phenolics per test sample. The punctual concentration series were as follows: 1, 5, 10, 25, 35, 50 µg GAE Eq in the case of strawberry fruit extract; 1, 5, 10, 25, 35, 50 µg pure compound (reference substance/r.s. solved in 50% ethanol) in the case of chalcones’ series; 1, 5, 10, 25, 35, 50 µg GAE Eq + r.s. (1:1, *w*/*w*) in the case of combined series. The control series were as follows: the negative control series 1 consisted of cells treated with Caco-2 growth medium, and the negative control series 2 consisted of cells treated with the solvent used for test samples (40% ethanol, following the dilution series of test samples). Since the two negative control series showed very similar values along the two MTS assays (STDEV < 0.05) and dilution series, the results in Figure 4, Figure 5 and Figure 6 are computed against the cumulative negative control series. More detailed data on the test sample preparation and the dilution series made can be found in Section 4.1, Section 4.2 and Section 4.4.

The statistical significance of the results (triplicate samples) was determined by Student *t*-test. Accordingly, the notation (*) means results without statistical significance (*p* > 0.05); the notation (**) means results with statistical significance (0.05 < *p* < 0.01); and the notation (***) means results with high statistical significance (*p* < 0.01).

Figure 4 presents the results of the MTS cytotoxicity assay after 24h (Figure 4a) and 48h (Figure 4b) of Caco-2 cells exposure to the seven test samples and dilution series. Particularly, the results on the strawberry extract series (S), chalcone derivatives series (phloridzin/Phd, phloretin/Phl, 4-methyl chalcone/4MeCh) and the three combinations between the strawberry extract and chalcone derivative series (S-Phd), (S-Phl) and (S-4MeCh), against the cumulative negative control series, mean values of the triplicates (*n* = 3).

Figure 4a,b, the results of the cytotoxicity assay, show the stimulatory effect upon the viability of the intestinal cells of strawberry extract (S), of phloretin (Phl) and of their combination (S-Phl), compared to the inhibitory effect of the 4-methylchalcone, alone (4-MeCh) and combined with strawberry extract (S-4MeCh). Phloridzin (Phd) and its combination with strawberry extract (S-Phd) appear as essentially inactive.

Particularly, after 24 h of cell exposure (Figure 4a), the 40% ethanolic extract from strawberries (S) enhanced the viability of the Caco-2 cells across the entire concentration range tested (1–50 µg GAE/sample), measuring 8 to 43% viability increased efficacy over the negative control series (red line); the maximum stimulatory efficacy was computed at 25 µg GAE/sample. Phloretin (Phl) series also showed stimulatory effects from 5 to 25 µg r.s./sample (up to 46%), while S-Phl showed stimulatory effects comparable to those of Phl alone (42%) at only 5 µg (GAE + r.s.)/sample. This result suggests that phloretin aglycone might be the dominant factor in this combination. The 4-MeCh alone indicated an inhibitory effect across the entire concentration range tested reaching -67% cell viability inhibition at the maximum concentration in the study (50 µg/sample); the combination with S (S-4MeCh) indicated cell viability inhibitory effects up to −59% (50 µg/sample), confirming the proliferative power of the strawberry ethanolic extract.

After 48 h of cell exposure (Figure 4b), the stimulatory effects of the S and Phl decreased and positioned closely to the negative control series; the combination S-Phl indicated fine stimulatory effects (+16%) at 5–10 µg (GAE + r.s.) and moderate inhibitory effects (−18%) in the interval between 35 and 50 µg (GAE + r.s.). The effects of Phd and S-Phd were consistent with those noticed at 24 h. The effects of 4-MeCh and S-4MeCh also were comparable, achieving −62% and −66% cell viability inhibitory activities, respectively.

Together, the cytotoxicity study essentially indicates that both, strawberry extract and phloretin are highly active vegetal products, and they can induce both stimulatory and inhibitory effects upon the viability of the intestinal cells in humans. Given these, the question that must be asked is the possibility of pro-tumor effects; studying cytotoxicity versus anti-proliferative effects (cell viability percents) at identical concentrations and study conditions (incubation time), the results were refined.

Figure 5 presents the cumulative data on the three pure compounds, cytotoxicity (Cy) versus anti-proliferative (AP) effects, after 48 h of Caco-2 cell exposure to the three pure compounds, also in comparison with the negative control series (red line). The pure compounds series are as follows: Phd-Cy48 and Phd-AP48 for phloridzin, Phl-Cy48 and Phl-AP48 for phloretin and 4MeCh-Cy48 and 4-MeCh-AP48 for 4-methylchalcone.

Figure 5 clearly shows the lack of activity of phloridzin glycoside. Phloretin aglycone indicated fine stimulatory effects from 1 to 35 µg r.s./sample and moderate inhibitory effects at 50 µg r.s./sample in cytotoxicity assay (Phl-Cy48), while in the anti-proliferative assay, (Phl-APp48) indicated inhibitory effects along the entire interval tested; the highest inhibitory effects at the maximum concentration level in the study (50 µg r.s./sample) achieved −34% Caco-2 cell viability inhibition. The 4-methylchalcone derivative indicated a consistent inhibitory activity, up to −62% in cytotoxicity assay (4-MeCh-Cy48) and up to −43% in anti-proliferative assay (4-MeCh-Ap48), respectively.

Similarly, Figure 6 presents the cumulative data on the strawberry ethanolic extract (S) and its combinations with the three pure compounds, cytotoxicity (Cy) versus anti-proliferative (AP) after 48 h of cell exposure, in comparison with the cumulative negative control series (the red line). The tested series are as follows: strawberry extract (S-Cy48 vs. S-AP48) series, strawberry extract combined with phloridzin (SPhd-Cy48 vs. SPhd-AP48), strawberry extract combined with phloretin (SPhl-Cy48 vs. SPhl-AP48) and strawberry extract combined with 4-methylchalcone (S4MeCh-Cy48 vs. S4MeCh-AP48).

Figure 6, particularly the cytotoxicity assay, suggests the ability of strawberry ethanolic extract (S-Cy48) to induce stimulatory effects upon the viability of Caco2 cells in the interval from 1 to 5 µg GAE/sample, and inhibitory effects at concentrations greater than these (the maximum inhibitory potency at 50 µg GAE per sample achieved −17% cell viability inhibition); in the anti-proliferative assay, the strawberry extract (S-AP48) indicated the ability to decrease the viability of the Caco-2 cells, from 2 to 18% magnitude, depending on the sample concentration tested. Strawberry ethanolic extract in combination with phloretin in cytotoxicity assay (SPhl-Cy48) revealed the potentiality of stimulatory effects from 5 to 10 µg GAE + r.s. per sample (+16%) and inhibitory effects (−18%) from 35 to 50 µg GAE + r.s. per sample; in the anti-proliferative assay, the effects were inhibitory (−20%) alongside the entire concentration range tested (SPhl-AP48). Strawberry ethanolic extract in combination with phloridzin (SPhd-Cy48 vs. SPhd-AP48) has proved ineffective in both assays, except for some slightly inhibitory effects noticed at the maximum concentration level in the study. 4-methylchalcone derivative and its combination with strawberry ethanolic extract (S4MeCh-Cy48 vs. S4MeCh-AP48) indicated identical inhibitory effects: −66% vs. −62% in cytotoxicity assay, and −43% vs. −42% in anti-proliferative assay.

Altogether, neither phloridzin nor 4-methylchalcone appears to improve the outcomes of strawberry ethanolic extract, and vice versa. Differently, strawberry extract combined with phloretin allowed both maximum cell viability stimulatory effects in the cytotoxicity assay (meaning cytoprotective activity) and maximum cell viability inhibitory effects in the anti-proliferative assay (meaning anti-tumor activity) at a minimal concentration level (5–10 µg) in the sample, practically revealing the advantage of using this combination for obtaining protective and guardian effects on the intestinal cells.

By comparison, similar MTT assays on human dermal fibroblast (HDF), on RAW macrophages and on hepatocellular carcinoma (HepG2) exposed to strawberry extracts in dilution series of 25, 50, 100, 250, 500, 1000, 2500, 5000, 7500 and 10,000 μg/mL test sample for 24, 48 and 72 h revealed the following results: no cytotoxic effects after 24 h of incubation in all cell lines; after 48 h of cell exposure, the cytotoxic effects occurred from 5000 μg/mL for HDF, from 7500 μg/mL for RAW macrophages and at the maximum concentration level, 1000 μg/mL, for HepG2; after 72 h of cell exposure, HepG2 indicated cytotoxic effects starting from 250 μg/mL, while HDF and RAW cells indicated a similar magnitude of cytotoxic effects at a 10-fold higher concentration than 48 h. Altogether, it was also concluded that the effects of strawberry extracts in vitro depend on the type of cells used, on the incubation time and the concentration samples used [101]. In consensus, studies on the total acetone extracts from strawberry fruits (variety Albion) reconstituted in 50% ethanol [102] indicated the ability of the concentration samples from 10 to 40 μg GAE/mL sample to induce both, stimulatory and inhibitory effects upon the viability of the Caco-2 cells, also depending on the incubation time and the concentration sample used. Furthermore, studies on the Maehyang cultivar (SE) indicated the ability of dry methanolic extracts from strawberries to inhibit the growth of the human colon (HCT-116), lung (A549), stomach (SNU-638) and fibrosarcoma (HT-1080) cancer cells; studies were carried out by sulforhodamine B/SRB assay in the interval from 0 to 10,000 µg SE/mL sample, and the results were evaluated after 72 h of cell exposure to the test extracts [103]. Other studies aimed to investigate the relationship between the antioxidant activity and the anti-proliferative activity of the strawberry extracts from eight current cultivars [104] also indicated augmented anti-proliferative effects upon the human liver cancer cell line HepG(2), especially in the case of Earliglow cultivar, but no relationship between the anti-proliferative activity and the antioxidant content. Finally, studies on a polyphenol-rich extract from strawberries indicated antitumor effects on both human breast (HT29) and human colon cancer (MCF-7) cell lines [105]; the inhibitory effect for the highest concentration in the study ranged in the interval 41–63% for HT29 and 26–56% for MCF-7. Very interestingly, the extracts from the organically grown fruits presented a higher anti-proliferative activity than the conventionally grown, for both types of cells.

## 3. Discussion

As exogenous compounds, the effects of the plant molecules in the human body depend heavily on their interactions with the cytochrome P450 (CYP) enzymes. Simultaneously, the compounds themselves can affect CYP function. Proving these, in silico analyses of the six small vegetal compounds under study emphasized their ability to interfere either with the activity of the cytochrome P450 oxidase system, or with the activity of the P-glycoprotein (P-gp) transporter in humans.

As is well known, the cytochrome P450 oxidase system is a protein complex comprising at least 57 monooxygenase-type enzymes (CYP) embedded in the endoplasmic reticulum of the cells [53,54]. CYP enzymes catalyze oxidation, reduction, desaturation, isomerization and hydrolysis reactions of both, exogenous and endogenous substrates. Altogether, CYP enzymes ensure Phase I metabolism reactions by adding new functional groups that make the exogenous molecules more polar and chemically reactive. Phase II metabolism reactions consist of conjugation processes via the attachment of an ionized group in the series of glutathione, methyl, sulfate, glycine, or acetyl to the molecules, rendering them much more soluble for excretion [106,107]. The glucuronidation process (via UDP-glucuronosyltransferase/UGT activity), also occurring at the level of the microsomal enzyme system in the liver, is the most important detoxification phase II reaction of xeno/endobiotics in humans [108].

Consequently, plant compounds influencing the activity of CYP enzymes may affect the general metabolism and the pharmacokinetics of drugs and other xenobiotics in humans. This way, in silico computation on the six test compounds indicated the ability of phloretin to inhibit three major CYP isoforms: CYP1A2, CYP2C9 and CYP3A4. 4-methylchalcone and hexadecanoic acid revealed the ability to inhibit CYP2C19 and CYP2D6, and CYP1A2 and CYP2C9 enzymes, respectively. In contrast, 2,3-dihydro -3,5-dihydroxy-6-methyl-4H-pyran-4-one and 2-pyrrolidinone 5-(cyclohexylmethyl) showed no interference with CYP enzymes and P-gp, while the phloridzin glycoside also does not appear to inhibit CYP enzymes but it could act as a P-gp substrate [77,109,110,111,112,113,114].

Regarding the projection of these inhibitory effects at the level of general metabolism in humans, CYP1A2 isoform represents approx. 3% of all CYP proteins in humans [55,56,57,58,59]. CYP1A2 enzyme is inducible in the liver, lung, pancreas, gastrointestinal tract and brain tissues, being particularly involved in the transformation of polycyclic aromatic hydrocarbons (PAHs), as those found in the cigarette and grilled meat smoke; CYP1A2 is also fully proven in the transformation of the dietary heterocyclic amines to their carcinogens, therefore is of major importance for liver function and human health. Other examples of CYP1A2 substrates are caffeine and theophylline, aflatoxin B1, acetaminophen and the pungent compounds from cruciferous; overall, it is estimated that the CYP1A2 isoform is involved in the metabolism of over 100 exogenous substrates, including antipsychotics and antithrombotic drugs, but also of estrogens [55,56,57,58,59].

CYP2C9 is another very important isoform representing up to 18% of the total cytochrome P450 protein complex in the human liver [60,61,62]. CTP2C9 appears to contribute to the metabolism of several drugs with crucial importance for humans; for example, the nonsteroidal anti-inflammatory drugs (NSAIDs) and the anticonvulsant, antidepressant and anticoagulant drugs, including those with low therapeutic index in humans [62] able to interfere with the activity of other drugs. Carbamazepine, ethanol and phenobarbitone are other examples of substrates for CYP2C9 involving the activity of the nervous system [63], therefore its inhibition is of high risk in respective patients.

CYP3A4 is the major isoform of CYP3A subfamily in adults, being responsible for most of the metabolic processes from the liver and intestines in humans; CYP3A can catalyze reactions of oxidation, hydroxylation, N-dealkylation, N-oxidation, N-hydroxylation, O/S-demethylation, sulfoxide formation and oxidative deamination of the exo/endogenous compounds too; cholesterol, bile acids, hormones, arachidonic acid and vitamin D are several examples of CYP3A substrates [64,65,66,67,68,69].

CYP2C subfamily enzymes also act on drugs of major importance for humans; it is thought to catalyze the transformation of at least 10% of the drugs used in clinical practice [70]. The proton pump inhibitors, hypnotic, sedative and antiepileptic drugs, antiplatelet drugs, antimalarial and antiretroviral drugs, progesterone, estrogen and antiandrogen drugs—all these are CYP2C substrates [71,72]. Much more, data show that the polymorphisms of *CYP2C19* play a “major role in inter-individual variability in drug response, drug-xenobiotic interactions, and in cancer susceptibility” [72].

CYP2D6 isoform is described as mainly found in the liver and in the areas of the central nervous system, being responsible for the metabolism and elimination of around 25% of the current drugs; this enzyme isoform catalyzes hydroxylation, demethylation and dealkylation processes, both in xenobiotic and endobiotic substrates, being also responsible for the poor or the ultra-rapid elimination of the drugs, therefore also playing a role in drug–drug interaction [73,74,75,76]. Furthermore, at the brain level, the CYP2D6 enzyme is responsible for the metabolism of serotonin, 5-hydroxytryptamine and other neuroactive steroids and amine neuromodulators [76]; therefore, its functionality is also of high importance for the function of the brain and neuronal system in humans.

Furthermore, phloridzin derivative comes out as interfering with the activity of P-glycoprotein (P-gp) transporter in humans. As is also well known, these membrane proteins are found throughout the entire human body, mostly at the apical pole of the epithelial cells with the excretory role (such as those found in the bile ductules, kidney tubules, colon and small intestines), and the level of the endothelial cells from blood vessels of the brain [77,109,110,111,112,113,114]. P-gp proteins act either for removing the absorbed xenobiotics from the intestine back to the gut lumen, or for accelerating xenobiotics’ transformation in the liver and kidney for the final purpose of being excreted into bile and urine, and also to protect the integrity of the blood–brain barrier [109,110,111,112]. Altogether, these proteins protect against the harmful effects of the oral xenobiotics by limiting their cellular uptake and distribution in the body, and by accelerating their clearance via excretory systems [77,109]. Regarding cancer disease, studies revealed that the P-gp proteins are over-expressed at the surface of most of the neoplastic cells, and thus they can block the internalization of the chemotherapeutics in cells being therefore involved in the multidrug resistance in humans [109,110,111,112]. This way, the P-gp modulators can modify the absorption, bioavailability and retention time of the oral xenobiotics, the cellular uptake of the drugs from the blood to the brain, as well as the internalization of the chemotherapeutics into the cancer cells; therefore, the P-gp substrates are of real concern and interest for managing the drug delivery and effectiveness in cancer disease.

On the other side, the three chalcone derivatives under studies displayed certain inhibitory potential against three protein–disease molecules involved in tumorigenesis and intestinal malignancy in humans, specifically on Bcl-2, TNKS1 and COX-2 targets.

Proving these, as is well known, the B-cell lymphoma family proteins (Bcl-2) govern the intrinsic apoptosis via mitochondrial outer membrane permeabilization in cells. The programmed cell death is essential for maintaining the homeostasis in tissues essentially occurring through the mitochondrial outer membrane permeabilization (MOMP) controlling, to finally activate the executors of the apoptosis process namely caspases [115,116,117,118]. The *BCL* gene activation is highly associated with B-cell chronic lymphocytic leukemia, but it also occurs in many other human malignancies. Since the balance between proliferation and apoptosis is particularly important in the case of tissues with high cellular turnover, this fact explains the crucial role of Bcl-2 in maintaining the homeostasis of the gastro-intestinal lining, particularly at the level of colon tissue abounding in intestinal stem cells (ISCs) [118,119,120,121]. Proving these, aiming to establish if and when the *BCL-2* is activated during colorectal cancer (CRC) and the relationship with the proteasome and tumor suppressor protein p53, very comprehensive studies concluded that “an inverse correlation was found between bcl-2 and p53 expression in adenomas” and the “percentage of bcl-2-positive cells were significantly more likely to have low rates of spontaneous apoptosis, than those cancers with low or absent bcl-2 expression”; these findings suggested that the abnormal activation of the *BCL-2* “appears to be an early event in colorectal tumorigenesis that can inhibit apoptosis in vivo and may facilitate tumor progression” [120]. It was concluded that the *BCL-2* genes are involved not only in the tumor initiation and tumor progression in patients with CRC, but also in the resistance to chemotherapy; furthermore, the abnormalities of BCL-2/Bcl-2 expression and activity were also related to neurodegenerative disorders, ischemia and autoimmune diseases [118,119,120,121]. These findings sustain the utility of Bcl-2 inhibitors in intestinal malignancy; several examples of drugs targeting Bcl-2 and solid tumors are Obatoclaxmesylate, Navitoclax, Palcitoclax, Gossypol, Apogosypol, Venetoclax and Antimycin [122].

Tankyrases (TNKS) are other protein–disease molecular targets, particularly important in intestinal malignancy. The human tankyrase-1 (TNKS1 or PARP5A) as well as the human tankyrase-2 (TNKS2 or PARP5B) are polymerase-type enzymes that control a multitude of cellular processes through the process of poly-ADP-ribosylation (PAR). As stated, poly-ADP-ribosylation is an essential signaling mechanism and a reversible post-translational modification that plays a significant role in many cellular processes; the Wnt signaling, as one of the key cascades regulating the cell development and stemness, is an example of cellular processes controlled by poly-ADP-ribosylation process assisted by tankyrases [123,124,125,126,127,128]. Briefly, poly-ADP-ribosylation controlls the activity of Axin, while Axin controls the activation of β-catenin, with a crucial role in carcinogenesis and malignancy. Specifically, the stimulation of the Wnt pathway results in β-catenin accumulation, which subsequently interacts with T-cell factor/lymphocyte enhancing–binding factor, thus activating the transcriptional activity in cells. Thus, each one failing in Wnt signaling can promote tumorigenesis and tumor progression in cells, but Axin appears to be the primary limiting component of this pathway [126,127,128,129]. Based on the observation that the inhibition of PAR formation or degradation can selectively eliminate cancer cells with specific DNA repair defects, and can enhance radiation or chemotherapy response in humans, the poly-ADP-ribose (PAR)-binding domains (PBDs) in tankyrases emerged as feasible targets for cancer therapy [130]. Studies have indeed proved that the tankyrases are particularly important in intestinal malignancy; the tankyrases’ inhibitors suppressed both, the Wnt signaling and the tumor growth in APC-mutant colorectal tumors [131,132,133,134,135,136,137,138], also decreasing the multi-drug resistance in TANKS-over-expressing colorectal cancer [138]. The existing inhibitors (e.g., olaparib, rucaparib, niraparib and talazoparib) are described with high effectiveness and reduced adverse effects on breast, ovarian and prostate tumors [136,137]. Promoting the degradation of the tankyrases is seen as another strategy to inhibit the Wnt/*β*-catenin pathway and to avoid the multi-drug resistance in CRC [139,140].

Cyclooxygenase-2 (COX-2) also is a protein–disease and a molecular target with high importance in tumorigenesis and intestinal malignancy in humans [141,142,143,144]. As is well known, COX-2 and COX-1 are two isoforms of prostaglandin H2 synthase (PGH) which assist the first step in the oxidation of arachidonic acid to prostanoids; the two isoforms have similar amino acid sequences, similar three-dimensional structure, similar active sites (valine–isoleucine substitutions are described at two positions only) and similar substrates, therefore lead to the same final products. COX-1 is a constitutive enzyme present in most human cells, either under physiological conditions or under disease conditions, while COX-2 is inducible and, therefore normally absent in human cells, but synthesized in large quantities within just a few hours of the onset of diseases involving the inflammatory process; large quantities of COX-2 are produced in chronic inflammation and cancer [145,146,147,148,149,150,151,152,153,154,155,156,157]. The conventional inhibitors (AINS or NSAIDs) decline both isoforms, while the selective inhibitors (e.g., rofecoxib and celecoxib) act on COX-2 isoform only [151,152,153]; this selectivity is based on a phylogenetic feature (COX-2 is more primitive than COX-1), rendering in small differences at the level of amino acids in the binding pocket [151]. COX-2 inhibitors are stated to have no effect on gastric mucosal prostaglandin synthesis, to cause no acute injury, and to no chronic ulceration compared to placebo; at the same time, they were demonstrated as chemopreventive in clinical trials [153,154,155,156,157], particularly in patients with familial adenomatous polyposis (FAP) [157]. Also, the COX-2 over-expression was specifically related to colorectal cancer (CRC) progression in humans, with a poor life prognosis in cancer disease and an increased probability of gaining drug resistance [154,155,156,157,158,159]. Thus, the use of the selective COX-2 inhibitors was proved with multiple benefits; e.g., there can be increased efficiency of the antitumoral drugs (the tamoxifen benefits were more augmented in the patients with breast cancer with over-expression of COX-2 treated with COX-2 inhibitors, compared to those who did not receive) [160], there can be reduced the toxicity of the chemotherapeutics, and there can be increased the radiosensitivity of the tumor tissues [157]. Hence, the major interest is in finding new selective inhibitors for COX-2 [157,158,159,160,161,162,163].

Furthermore, the reviewed data [23,24,164] compiling studies on various human cancer cell lines and carcinomas from breast, liver, lung, gastric, oral, esophageal, colon, cervical and prostrate indicated cumulative tumor cell growth, adhesion, migration, metastasizing and angiogenesis inhibitory effects, while the apoptosis’ modulation emerged as a primary anti-tumor mechanism [23,135,164]. Proving these, exploratory studies on the oral cancer cell lines YD-9 and CA9-22 indicated phloretin up-regulation effects upon the apoptotic proteins BAX, cytochrome c, PARP, caspases 3 and caspase 9 alongside the apoptotic activating factor (APAF), concomitantly with phloretin down-regulation effects upon the expression of the anti-apoptotic protein Bcl-2 [165]. Studies on the human gastric cancer cell line (BGC823) indicated the ability of phloretin to inhibit both, cell growth and cell invasion in vitro; specifically, it was proved the ability of phloretin to produce the cell cycle arrest in the G2/M phase at the same time to inhibit the c-Jun N-terminal kinase (JNK) signaling pathway in cells [166]. Other studies on human breast carcinoma exposed to a ruthenium–phloretin complex revealed the ability of phloretin to induce stimulatory effects upon the apoptosis process (via p21-, Cyt-C-, caspase 9-, cleaved caspase 3- and Bax-mediated signaling pathways) at concomitantly with inhibitory effects upon the activity of anti-apoptotic protein Bcl-2 [167]. The ability of phloretin to inhibit the activity of type 2 glucose transporter (GLUT2) in cells has also been related to apoptosis and anti-metastasizing effects; e.g., glucose deprivation induced by phloretin induced G0/G1 cell cycle arrest, ATP depletion and the activation of the death pathway in mitochondria [168]. Studies on HepG2 cells exposed to phloretin also indicated an increase in G0/G1 and G2/M cell phases [169], while on hepatoma cells indicated G1 cell phase arrest effects, suggesting the possibility that the phloretin interferes with phenolics from DNA helix [168].

Similarly, the compound xanthohumol, a chalcone isolated from *Humulus lupulus,* incubated with breast cancer cell line MCF-7/ADR [170] indicated the ability to suppress the estrogen signaling pathways (via tumor suppressor protein prohibitin2 modulation) and the expression of the anti-apoptosis proteins, at the same time to induce cell cycle perturbation and to sensitize the cancer cells to radiation treatment. Cardamonin, a chalcone derivative from Alpina Katsumadai, tested on hepatocellular carcinoma (HCC) and HepG2 cells [171,172], showed effects upon the intrinsic and the extrinsic apoptosis pathways in mitochondria; the anti-tumor effect occurred via the stimulation of the intracellular generation of radical oxidative species/ROS) concomitantly with the inhibition of the nuclear factor Nf-kB pathway and other important apoptosis regulator proteins in cells (e.g., the cell nuclear antigen/PCNA, the antigen Ki-67 nuclear protein and the B cell-lymphoma proteins Bcl-2 and Bax). Studies on the human pulmonary carcinoma cell line A549 [173] exposed to helichrysetin (a chalcone derivative from *Helichrysum odoratissimum*) also concluded a mitochondrial-induced apoptosis mechanism via DNA fragmentation and cell cycle arrest in the S phase.

In regards to superior clinical evaluations, very recent reviewed data [164] counted 12 chalcone derivatives passing extended in vitro and in vivo studies: naringenin, phloretin, xanthohumol, licochalcone A and licochalcone B, butein, isoliquiritigenin, helichristen, cardamonin, flavokawain A, xantoangelol and 4-hydroxyderrecin. Of these, four compounds were allowed for the phase I clinical studies (naringenin, phloretin, xanthohumol, licochalcone A) and only two compounds for phase II clinical studies (xanthohumol, licochalcone A for chronic venous disease in women). There are no data regarding the results obtained in clinical testing of phloretin chalcone; the specialized platform ClinicalTrials.gov [174] indicates “Exploratory assessment of the effects of cyclo3fort (containing Ruscus Extract, Hesperidin, Methyl Chacone and Ascorbic Acid) or micronised purified flavonoid fraction on vascular parameters and biomarkers in women suffering from chronic venous disease” only. Summing all, the fact that such a large number of chalcone derivatives are of clinical interest indicates the huge curative potential of this flavonoid subclass, and their major human health benefits.

## 4. Materials and Methods

### 4.1. Preparation of the Strawberry Extract

Strawberry fruits, the variety “Albion”, were selected for homogenous aspects avoiding immature or wounded fruits. The selected fruits (approx. 550 g) were washed in tap water, and then rinsed in distilled water after that they were placed on the cellulose paper. The dried fruits were then smashed to a homogeneous paste in a ceramic mortar; 500.00 g of fruit paste was placed in an Erlenmeyer flask with a ground stopper (1000 mL total capacity). A total of 500 mL of ethanol of concentration 96% was added to the fruit paste. The mixture was placed in a dark place at 18–20 °C and shaken vigorously for 10–15 min daily, 12 days total time. The final macerate was filtered on a double paper filter (Roth 1506, Karlsruhe, Germany); the resulting ethanolic extract was measured (mL) and analyzed for polyphenols and total extractable contents. The ethanolic extract was concentrated in a Buchi Rotavapor to a *spiss* product; the *spiss* product was solubilized in 40% ethanol (on the ultrasound bath at 40 °C) to ensure a precise content of 12.5 mg total phenolics (gallic acid equivalents, GAE Eq) per 10 mL total extract (meaning stock solution of 125 mg%). The resulting (standardized) extract (codified S) was used to prepare a dilution series for pharmacological studies and GC-MS analysis.

### 4.2. Preparation of the Reference Substances

Phloretin (by Molekula GmbH, München, Germany), phlorizin hydrate (by TCI, Tokio Chemical Industry, Tokio, Japan) and 4-methylchalcone (by Sigma Aldrich, Merck, Darmstadt, Germany) reference substances (r.s.) were prepared as identical solutions of 12.5 mg pure compound solved in 10 mL of 50% ethanol solvent (meaning stock solution of 125 mg%); the resulting suspensions were maintained at ultrasounds, 15 min at 40 °C, for complete dissolution; they were codified as Phd (Phloridzin), Phl (Phloretin) and 4-MeCh (4-Methylchalcone).

### 4.3. Analytical Studies on the Strawberry Extracts

The total polyphenols content was assessed by the Folin–Ciocalteau method, as described in Romanian Pharmacopoeia, FRX. [175]; the measurements were carried out using UV/Vis Hélios γ (Thermo Electron Corporation, Waltham, MA, USA) spectrophotometer. The results were estimated by comparison with the gallic acid (reference substance) calibration curve (R^2^ = 0.979, n = 3), and expressed as mg [GAE] per 100 mL extract.

The total extractable content was also assessed by FRX recommendations [175]: three series of 50 mL strawberry fruit extract, each one was placed in a round-bottomed flask weighed as two decimals; the three samples were then evaporated at rotavapor to residue, then each one residue was solved twice with 50 mL of absolute ethanol and re-evaporated to dryness. The flasks with the (three) residues were placed in a desiccator until the next day, and then weighed at two decimals. The quantity (mean value, n = 3) represents the amount of the extractable substances in a test extract. Particularly, gravimetrical measurements in the case of the strawberry extract indicated a content of 2.8 g (±5%) total extractable per 100 mL test extract.

The GC-MS analysis was driven by CLARUS 500 (PerkinElmerR, Waltham, MA, USA) apparatus using a capillary column made of fused silica (L = 60m and di = 0.32 mm) and a stationary phase composed of dimethyl (95%) and diphenyl (5%) siloxane solvents, thus allowing a film thickness of 0.25 μm. The apparatus is equipped with TurboMass software v5.1 for the on-line management of the instrument processing the GC-MS data. The working conditions were as follows: the temperature at the injector −250 °C; initial temperature in the oven −100 °C; the heating range, from 10 °C to 200 °C (2 min); the heating rate, 4 °C per minute up to 280 °C, 8 min total time; the flow rate—helium 1 mL/minute; the injected sample volume—2 μL. The conditions for the MS were as follows: 280 °C temperature at the GC-MS interface; 250 °C temperature in the ion source (EI); 70 eV electron energy; acquisition mode—SCAN. The GC-MS results are listed in Table 1.

### 4.4. In Silico Studies

In silico analyses comparing the physical-chemical, pharmacokinetic, medicinal chemistry and lead-likeness parameters alongside the bioavailability features and potential interferences with the activity of P-glycoprotein (P-gp) transporter and the cytochrome P450 oxidase system in humans were carried out by SwissADME free web tool operated by the Molecular Modelling Group of the University of Lausanne and the SIB Swiss Institute of Bioinformatics [39].

In silico molecular docking studies aiming to analyze the inhibitory potential of the three phloridzin derivatives upon the activity of the three protein–disease targets proved as being relevant in relation to the intestinal tumorigenesis (Bcl-2, TNKS1 and COX-2) were carried out by CLC Drug Discovery Workbench 2.4 software (QIAGEN, Aarhus, Denmark) [47]. Studies were made in comparison with their native ligands (validated inhibitors), retrieved from the Protein Data Bank (https://www.rcsb.org, accessed on 22 March 2025) [93]. The three ligands are as follows: the PDB ID:2O2F corresponding to the Bcl-2 target in complex to the native inhibitor namely the co-crystallized ligand LI0, a benzamide derivative such 4-(4-benzyl-4-methoxy piperidin-1-yl)-N-[(4-{[1,1-dimethyl-2-(phenylthio)ethyl] amino}-3-nitrophenyl)sulfonyl]benzamide [94]; the PDB ID: 4W6E corresponding to the TNKS1 target in complex to the native inhibitor namely co-crystallized 3J5A, a quinazoline derivative such 2-(4-{6-[(3S)-3,4-dimethylpiperazin-1-yl]-4-methylpyridin-3-yl}phenyl)-8-(hydroxymethyl) quinazolin-4(3H)-one [95]; the PDB ID: 1PXX corresponding to the COX-2 target in complex to the native ligand namely diclofenac, the 2-[2,6-dichlorophenyl)amino]benzeneacetic acid, respectively [96]. The molecular targets are first of all prepared for the docking analysis by removing the co-factors in the series of metals, small molecules and water, by protonation and by setup the binding site and the binding pocket at the specific bond legs (Å^2^). The ligands are also prepared by energy minimization using the Spartan’14 software program from Wavefunction, Inc., Irvine, CA, USA [97]. The molecular docking validation is made by re-docking the native ligands in comparison with the tested ligands. In the end, the punctual hydrogen bonds and the docking scores are compared, the native ligands in comparison with tested ligands, to obtain their scale of efficacy.

### 4.5. In Vitro Pharmacological Studies

Pharmacological studies were carried out on the Caco-2 cell line (ATCC, HTB-37) following the MTS cytotoxicity and anti-proliferative assays, as described by CellTiter 96AQueous One Solution Cell Proliferation Assay, Promega Corporation (Madison, WI, USA) [100]. Briefly, after reaching the cell confluence necessary for cytotoxicity (70%) and anti-proliferative (30%) assays, the cells are detached from the flask with Trypsin–EDTA and the resulting cell suspension is centrifuged at 2000 rpm for 5 min. The separated cells are re-suspended in Caco-2 growth medium, and then seeded in 96-well plates at a density of 8000 and 4000 cells per well in 200 μL culture medium, respectively. The cells in the well plates are incubated with 200 μL of test/control samples, following the dilution/concentration scale designed. In the current study, the 40% ethanolic extract from strawberry fruits (S) and the tree chalcone derivatives (Phd, Phl, 4-MeCh pure compounds) presenting as stock solutions of 125 mg% were prepared each one as six dilution series; punctually, 8, 40, 80, 200, 280 and 400 µL test extract/test chalcone solution were mixed with 1992, 1960, 1920, 1800, 1720 and 1600 µL Caco-2 growth medium, thus obtaining the dilution scale of 1, 5, 25, 35 and 50 µg GAE Eq/r.s. (*w*/*w*) per 200 μL test sample. In the specific case of their combination, first, the 1:1 combination between S and chalcone derivatives (Phd, Phl, 4-MeCh) was made, and after that, identical dilution series were prepared. The cells in the well plates labeled as negative control series 1 were treated with 200 μL Caco-2 growth medium supplemented with FBS (Merck Sigma-Aldrich, Saint Louis, MO, USA, distributor in Romania), while the cells in the well plates labeled as negative control series 2 were treated with 200 μL 40% ethanol following the dilution series for the test extracts (8, 40, 80, 200, 280 and 400 µL 40% ethanol were mixed with 1992, 1960, 1920, 1800, 1720 and 1600 µL Caco-2 growth medium, respectively). Each test/control/point series was prepared as triplicate samples. After 20 h and 44 h of the cell exposure to the test/control dilution samples, the culture medium was removed and the cells were exposed to MTS solution for another 2 h. At the end of the study, the plates were read at 492 nm (BMR-100 Microplate Reader, Boeco, Germany); the optical densities (O.D. at 492 nm) obtained were transformed in cell viability percentages (by computing the test series against the negative control series) and computed for their statistical significance (Student “*t*” test). The notation (*) means results without statistical significance and *p* > 0.05; the notation (**) means results with statistical significance and 0.05 < *p* < 0.01; the notation (***) means results with statistical significance and *p* < 0.01.

## 5. Conclusions

In silico SwisADME analysis indicated the ability of phloretin, 4-methylchalcone and hexadecanoic acid to inhibit the activity of the main cytochrome P450 (CYP) enzymes in humans; phloridzin showed the ability to act as P-gp inhibitor, while 2,3-dihydro-3,5-dihydroxy-6-methyl-4H-pyran-4-one and 2-pyrrolidinone 5-(cyclohexyl methyl) derivatives showed no interference either with CYP enzymes or P-gp transporter in humans. In silico CLC docking results indicated the ability of phloridzin, phloretin and 4-methylchalcone to act as moderate inhibitors of Bcl-2, TNKS1 and COX-2, three important protein disease targets involved in tumorigenesis.

In vitro studies, MTS cytotoxicity and anti-proliferative assays on Caco-2 cells, led to the following conclusions: the 40% ethanolic extract from strawberries presents a degree of uncertainty (therefore a risk) in relationship with the human intestinal tumor cells since revealed both stimulatory (up to +43%) and inhibitory potentiality (up to −17%) depending on the dose, the incubation time (24 h or 48 h) and the type of the assay performed (cytotoxicity or anti-proliferative). This stimulatory risk on the Caco-2 cells was even more increased in the previous studies on the total acetonic extracts from strawberry fruits reconstituted in 50% ethanol [102]. Phloretin (Phl) pure compound also indicated stimulatory effects (up to +46%) as well as inhibitory effects (up to −34%) depending on the dose and the type of assay performed. Phloridzin (Phd) pure compound indicated zero or very weak activity on the Caco-2 cells. The 4-methylchalcone (4-MeCh) pure compound indicated consistent inhibitory effects in both MTS assays, −62% vs. −42% cell viability inhibition, respectively.

Furthermore, strawberry ethanolic extract combined with phloretin acted as follows: at low concentrations, from 5 to 10 µg GAE + r.s. (1:1, *w*/*w*) per sample, the combination increased the Caco-2 viability in cytotoxicity assay (+16%) and inhibited the cell viability in the anti-proliferative assay (−20%); at higher concentrations, the effects in both MTS assays were inhibitory (−20%). Strawberry ethanolic extract combined with phloridzin indicated the lack of biological effects in both MTS assays. Strawberry ethanolic extract combined with 4-methylchalcone proved to be useless since the results were similar to those noticed in the case of the pure compound (−66% vs. −22% and −43% vs. −40%).

By in vitro–in silico data corroboration, the fact that phloridzin did not interact with the human intestinal cells in vitro suggests the inability of these cells to internalize, or to take up the signal of this compound from the medium; this lack of activity could be estimated from in silico studies, that indicated the low bioavailability of the phloridzin glycoside in GI. Differently, phloretin and 4-methylchalcone showed increased bioavailability in GI and also induced stimulatory and/or inhibitory effects upon the intestinal Caco-2 tumor cells in vitro. This way, in silico docking analysis should be directed toward the compounds proved with the ability to pass the GI barrier. However, the formulation methods today can overcome the intestinal barrier in humans, so that the compounds ineffective on the intestinal cells could be effective on other human cells.

Altogether, it can be concluded that phloretin might be used alone or in combination with strawberry ethanolic extract to support intestinal cell health in humans. Both variants showed a beneficial stimulatory effect in the cytotoxicity assay versus an inhibitory effect in the anti-proliferative assay; the combination between strawberry ethanolic extract and phloretin is as secure as the metabolite of the phloretin in humans, and 4-methylchalcone proved consistent cytotoxic and anti-proliferative effects. Supporting these, very recent studies [176] revealed that phloretin suppressed the intestinal inflammation and also preserved the epithelial tight junction integrity in in vitro gut inflammation model (developed by co-culture of Caco2 intestinal cells and RAW264.7 macrophage immune cells). Particularly, phloretin decreased the level of the reactive oxygen species (ROS), of lipopolysaccharides (LPS) and also of numerous inflammatory cytokines (e.g., interleukins IL8, IL1β, IL6 and tumor necrosis factor/TNFα) and several very active pro-inflammatory proteins (e.g., Nuclear factor NF-κB, inducible nitric oxide synthase/iNOS and Cyclooxygenase-2/Cox-2), while also maintaining the gut epithelial integrity by regulating the expression of tight junction proteins ZO1, occludin, Claudin1 and junctional adhesion molecules (JAM). Based on these data, the authors recommended the use of phloretin as a nutraceutical in preventing the occurrence of colitis and culmination of disease into colitis associated colorectal cancer.

Regarding the limitations in the present study, there cannot be explained the basis of dual stimulatory—inhibitory effects of the alcoholic extract from strawberry fruits noticed in both cytotoxicity and anti-proliferative assays, all the more so as these effects were transmitted in combination with phloretin. Also, further in vitro tests are needed, on human normal and human tumor liver cell lines, with the precise aim of evaluating the dynamic of cytoprotective—cytotoxic ratio along the concentration series. This is in the context in which in silico analysis also indicated the ability of phloretin to inhibit the activity of three major CYP enzymes (CYP1A2, CYP2C9, CYP3A4), while aging can reduce the hepatic volume and the blood flow diminishing CYP capacity by over 30% [53]. It must be noted that the five CYP enzymes under study cover over 70% of the enzymatic function of cytochrome P450 in humans. Finally, resulting as an effective inhibitor of the Caco-2 cell growth and viability in vitro, in both cytotoxicity and anti-proliferative MTS assays, the 4-methylchalcone derivative is a strong candidate for further studies, especially in combinations with validated anticancer drugs.

## Figures and Tables

**Figure 1 ijms-26-03492-f001:**
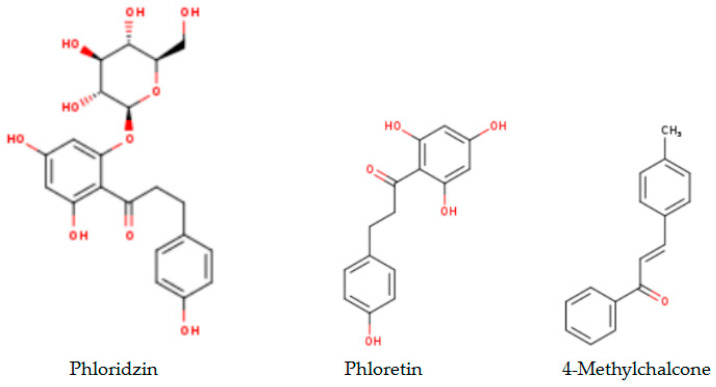
(**a**) The chemical structure of the phloridzin derivatives. (**b**) The chemical structure of the three small compounds identified by GC-MS analysis in 40% ethanolic extract from strawberries.

**Figure 2 ijms-26-03492-f002:**
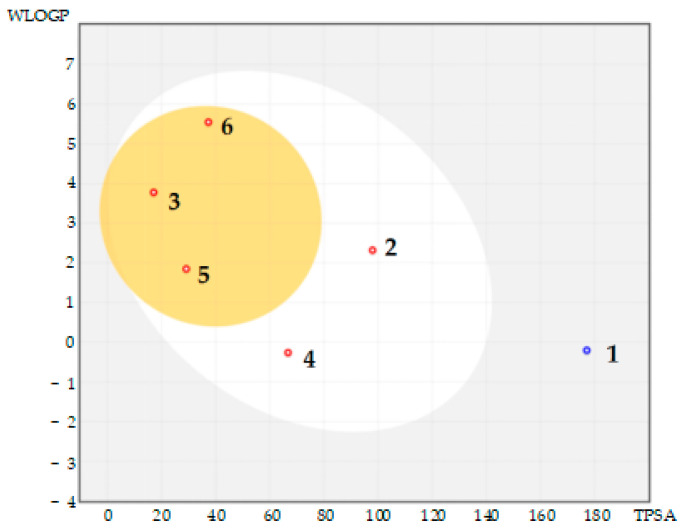
The BOILED-Egg representation (WLOGP versus TPSA, [41]) for phloridzin (1), phloretin (2), 4-methylchalcone (3) and 2,3-dihydro-3,5-dihydroxy-6-methyl-4H-pyran-4-one (4), 2-pyrrolidinone 5-(cyclohexylmethyl) (5) and hexadecanoic acid (6), respectively.

**Figure 3 ijms-26-03492-f003:**
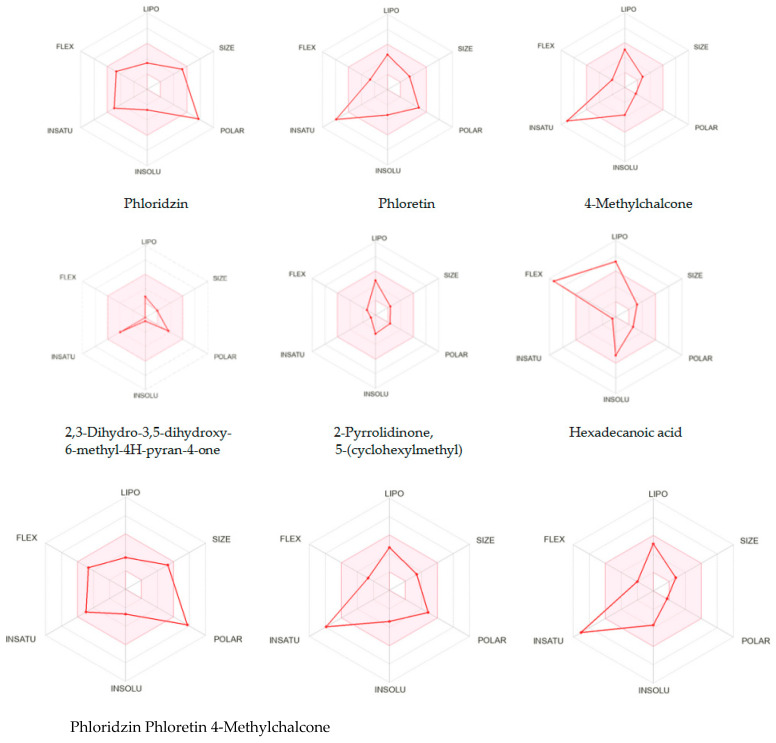
The SwisADME bioavailability radar of the six plant compounds under study.

**Figure 4 ijms-26-03492-f004:**
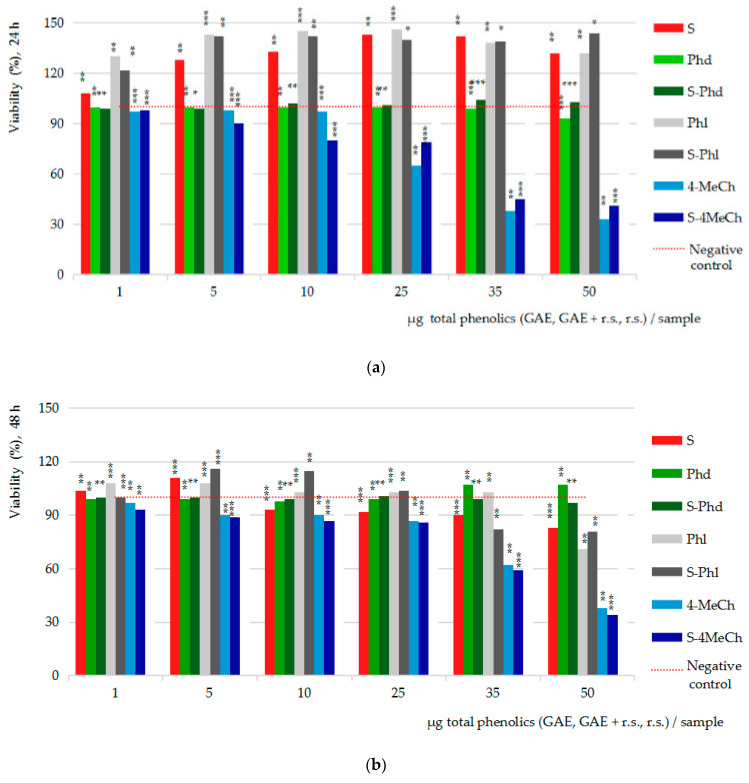
(**a**) MTS cytotoxicity assay results after 24 h of Caco-2 cell exposure to the seven test samples, the 40% ethanolic strawberry extract series (S), the three pure compounds (r.s.) phloridzin (Phl), phloretin (Phl) and 4-methylchalcone (4-MeCh) series, and their combined (S-Phd, S-Phl and S-4MeCh) series in rate 1:1 (GAE:r.s., *w*/*w*) by comparison with the negative control series, following the concentration series from 1 to 50 µg total active compounds (gallic acid derivates and pure chalcones, GAE:r.s) per 1 mL sample. The results are presented as viability percents (%) against the negative control series (the red line) and mean values for triplicate samples (*n* = 3), respectively. Notation (*) means results without statistical significance (*p* > 0.05); notation (**) means results with statistical significance (0.01 < *p* < 0.05); notation (***) means results with statistical significance (*p* < 0.01). (**b**) MTS cytotoxicity assay results after 48h of Caco-2 cell exposure to the seven test samples, the 40% ethanolic strawberry extract series (S), the three pure compounds (r.s.) phloridzin (Phl), phloretin (Phl) and 4-methylchalcone (4-MeCh) series, and their combined (S-Phd, S-Phl and S-4MeCh) series in rate 1:1 (GAE:r.s., *w*/*w*) by comparison with the negative control series, following the concentration series from 1 to 50 µg total active compounds (gallic acid derivates and pure chalcones, GAE:r.s) per 1 mL sample. The results are presented as viability percents (%) against the negative control series (the red line) and mean values for triplicate samples (*n* = 3), respectively. Notation (**) means results with statistical significance (0.01 < *p* < 0.05); notation (***) means results with statistical significance (*p* < 0.01).

**Figure 5 ijms-26-03492-f005:**
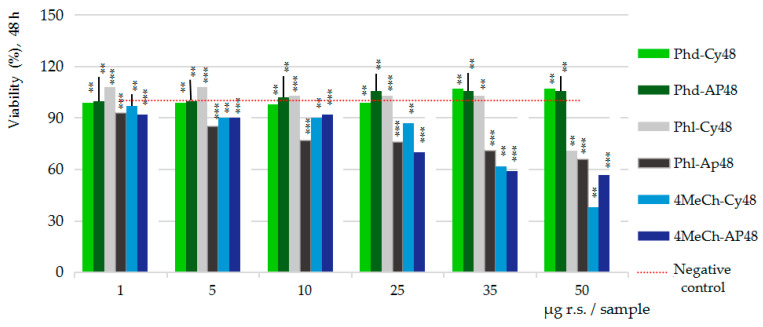
Cytotoxicity vs. anti-proliferative activity after 48 h of Caco-2 cell exposure to phloridzin (Phl), phloretin (Phl) and 4-methylchalcone (4-MeCh) pure compounds’ dilution series (reference substances/r.s. prepared in 50% ethanol), following the concentration series from 1 to 50 µg pure chalcone (r.s.) per 1 mL test sample. The results are presented as viability percents (%) against the negative control series (the red line) and mean values for triplicate samples (*n* = 3), respectively. Notation (**) means results with statistical significance (0.01 < *p* < 0.05); notation (***) means results with statistical significance (*p* < 0.01).

**Figure 6 ijms-26-03492-f006:**
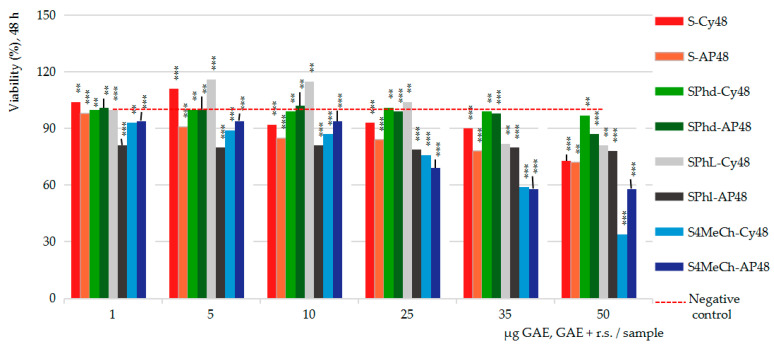
Cytotoxicity vs. anti-proliferative activity after 48 h of Caco-2 cell exposure to 40% ethanolic extract from strawberries (S-Cy48 vs S-AP48) and its combinations with phloridzin (SPhd-Cy48 vs. SPhd-AP48), phloretin (SPhl-Cy48 vs. SPhl-AP48) and 4-methylchalcone (S4MeCh-Cy48 vs. S4MeCh-AP48) pure compounds dilution series, following the concentration series from 1 to 50 µg total active compounds (gallic acid derivates and pure chalcone, GAE + r.s., 1:1, *w*/*w*) per 1 mL sample. The results are presented as viability percents (%) against the negative control series (the red line) and mean values for triplicate samples (*n* = 3), respectively. Notation (**) means results with statistical significance (0.01 < *p* < 0.05); notation (***) means results with statistical significance (*p* < 0.01).

**Table 1 ijms-26-03492-t001:** The GC-MS analysis of the 40% ethanolic extract from strawberry fruits.

Test Compound	MW	Formula	Peak Area	Retention Time (min)	Probability (%)	Concentration in Sample (%)
2,3-dihidro-3,5-dihidroxi- 6-metil-4H-piran-4-ona	144	C_6_H_8_O_4_	782,240,832	8.50	93.4	11.03
2-pyrrolidinone 5-(cyclohexylmethyl)	181	C_11_H_19_ON	138,499,424	12.54	95.5	1.95
Acid n-hexadecanoic	256	C_16_H_32_O_2_	44,022,120	19.87	93.7	0.62

**Table 2 ijms-26-03492-t002:** (**a**) In silico analysis of the phloridzin derivatives. (**b**) In silico analysis of the three small compounds identified in the 40% ethanolic extract from strawberry fruits.

(**a**)			
**Test Compound**	**Phloridzin**	**Phloretin**	**4-Methylchalcone**
Physical-chemical properties			
Formula	C_21_H_24_O_10_	C_15_H_14_O_5_	C_16_H_14_O
Molecular weight	436.41 g/mol	274.27 g/mol	222.28 g/mol
Num. heavy atoms	31	20	17
Num. arom. heavy atoms	12	12	12
Fraction Csp3	0.38	0.13	0.06
Num. rotatable bonds	7	4	3
Num. H-bond acceptors	10	5	1
Num. H-bond donors	7	4	0
Molar refractivity	106.14	74.02	71.21
TPSA	177.14 Å^2^	97.99 Å^2^	17.07 Å^2^
Lipophilicity			
Log *P*_o/w_ (iLOGP)	1.25	1.41	2.78
Log *P*_o/w_ (XLOGP3)	0.54	2.63	3.44
Log *P*_o/w_ (WLOGP)	−0.20	2.32	3.78
Log *P*_o/w_ (MLOGP)	−1.42	1.10	3.69
Log *P*_o/w_ (SILICOS-IT)	0.13	2.17	4.45
Consensus Log *P*_o/w_	0.06	1.93	3.63
Water solubility			
Log *S* (ESOL)	−2.71	−3.38	−3.71
Log *S* (Ali)	−3.83	−4.34	−3.48
Log *S* (SILICOS-IT)	−1.66	−3.37	−5.35
Pharmacokinetic			
GI absorption	Low	High	High
BBB permeant	No	No	Yes
P-gp substrate	**Yes**	No	No
CYP1A2 inhibitor	No	Yes	No
CYP2C19 inhibitor	No	No	Yes
CYP2C9 inhibitor	No	Yes	No
CYP2D6 inhibitor	No	No	Yes
CYP3A4 inhibitor	No	Yes	No
Medicinal Chemistry			
PAINS	0 alert	0 alert	0 alert
Brenk	0 alert	0 alert	1 alert: Michael_acceptor_1
Synthetic accessibility	4.93	1.88	2.53
Lead-likeness	No, 1 violation: MW > 350	Yes	No, 1 violation: MW < 250
Drug-likeness			
Lipinski	Yes, 1 violation: NH or OH > 5	Yes, 0 violation	Yes, 0 violation
Ghose	Yes	Yes	Yes
Veber	No, 1 violation: TPSA > 140	Yes	Yes
Egan	No, 1 violation: TPSA > 131.	Yes	Yes
Muegge	No, 2 violations:	Yes	No, 1 violation:
	TPSA > 150, H-don > 5		Heteroatoms < 2
Bioavailability score	0.55	0.55	0.55
(**b**)			
**Test Compound**	**2,3-Dihydro-3,5-dihydroxy-** **6-methyl-4H-pyran-4-one**	**2-Pyrrolidinone 5-(cyclohexylmethyl)**	**Hexadecanoic Acid**
Physical-chemical properties			
Formula	C_6_H_8_O_4_	C_11_H_19_NO	C_16_H_32_O_2_
Molecular weight	144.13 g/mol	181.27 g/mol	256.42 g/mol
Num. heavy atoms	10	13	18
Num. arom. heavy atoms	0	0	0
Fraction Csp3	0.50	0.91	0.94
Num. rotatable bonds	0	2	14
Num. H-bond acceptors	4	1	2
Num. H-bond donors	2	1	1
Molar refractivity	32.39	57.68	80.80
TPSA	66.76 Å^2^	29.10 Å^2^	37.30 Å^2^
Lipophilicity			
Log *P*_o/w_ (iLOGP)	1.19	2.39	3.85
Log *P*_o/w_ (XLOGP3)	−0.37	2.77	7.17
Log *P*_o/w_ (WLOGP)	−0.26	1.85	5.55
Log *P*_o/w_ (MLOGP)	−1.77	2.07	4.19
Log *P*_o/w_ (SILICOS-IT)	0.13	2.61	5.25
Consensus log *P*_o/w_	−0.22	2.34	5.20
Water solubility			
Log *S* (ESOL)	−0.50	−2.58	−5.02
Log *S* (Ali)	−0.57	−3.04	−7.77
Log *S* (SILICOS-IT)	0.15	−2.54	−5.31
Pharmacokinetic			
GI absorption	High	High	High
BBB permeant	No	Yes	Yes
P-gp substrate	No	No	No
CYP1A2 inhibitor	No	No	Yes
CYP2C19 inhibitor	No	No	No
CYP2C9 inhibitor	No	No	Yes
CYP2D6 inhibitor	No	No	No
CYP3A4 inhibitor	No	No	No
Medicinal chemistry			
PAINS	0 alert	0 alert	0 alert
Brenk	0 alert	0 alert	0 alert
Synthetic accessibility	3.60	1.99	2.31
Lead-likeness	No, 1 violation:	No, 1 violation:	No, 2 violations:
	MW < 250	MW < 250	Rotors > 7, XLOGP3 > 3.5
Drug-likeness			
Lipinski	Yes, 0 violation	Yes, 0 violation	Yes, 1 violation: MLOGP > 4.15
Ghose	No, 3 violations: MW < 160,	Yes	Yes
	MR < 40, no. atoms < 20		
Veber	Yes	Yes	No, 1 violation: Rotors > 10
Egan	Yes	Yes	Yes
Muegge	No, 1 violation:	No, 1 violation:	No, 1 violation:
	MW < 200	MW < 200	XLOGP3 > 5
Bioavailability score	0.85	0.55	0.85

**Table 3 ijms-26-03492-t003:** Molecular docking results on Bcl-2 target (PDB ID: 2O2F) [94].

Test Pairs	The Docking Image of the Test Ligand Interacting with the Amino Acid Residues in the Binding Site, Chain A	The Punctual Hydrogen Bonds: Å	Score/RMSD
2O2F in complex with co-crystallized LI0 A 1000	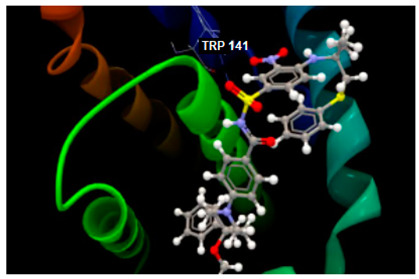	O sp^2^–N sp^2^ TRP141: 3.281	−82.97/2.02 [93]
2O2F in complex with phloridzin	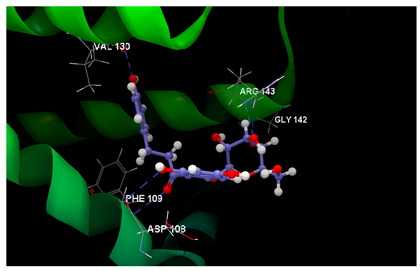	O7 sp^2^–N sp^2^ PHE109: 3.066 O6 sp^3^–O sp^2^ ASP108: 3.330 O2 sp^3^–N sp^2^ ARG143: 2.619 O3 sp^3^–O sp^2^ GLY142: 3.176 O3 sp^3^–N sp^2^ ARG143: 3.351 O9 sp^3^–O sp^2^ VAL130: 2.874	−62.85/0.09
2O2F in complex with phloretin	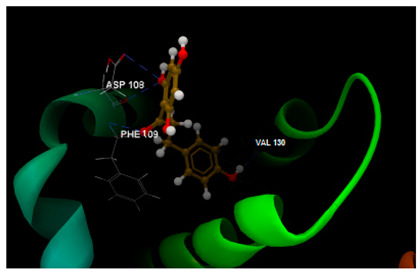	O0 sp^2^–N sp^2^ PHE109: 3.238 O4 sp^3^–O sp^2^ VAL130: 2.062 O1 sp^3^–O sp^2^ ASP108: 3.093 O1 sp^3^–O sp^2^ ASP108: 3.235	−51.97/0.15
2O2F in complex with 4-methylchalcone		no bonds	−52.09/0.10

**Table 4 ijms-26-03492-t004:** Molecular docking results on TNKS1 target (PDB ID: 4W6E, chain A) [95].

Test Pairs	The Docking Image of the Test Ligand Interacting with the Amino Acid Residues in the Binding Site	The Punctual Hydrogen Bonds: Å	Score/RMSD
4W6E in complex with co-crystallized 3J5A	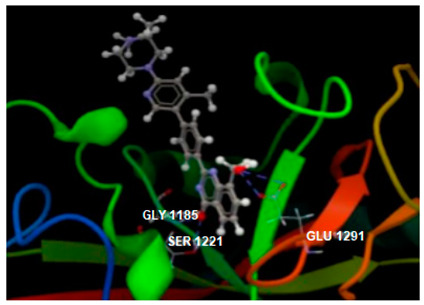	O1 sp^3^–O sp^2^ GLU1291: 3.393 O1 sp^3^–O sp^2^ GLU1291: 3.250 O sp^2^–O sp^3^ SER1221: 2.778 O sp^2^–N sp^2^ GLY1185: 2.895	−104.15/0.16 [52]
4W6E in complex with phloridzin	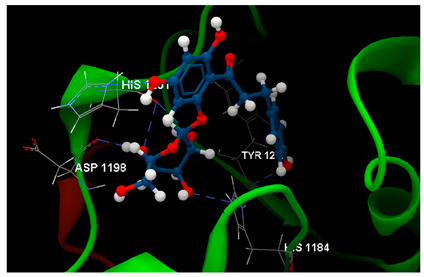	O9 sp^3^–Osp^2^ TYR1213: 2.977 O8 sp^3^–Nsp^2^ HIS1201: 2.688 O3 sp^3^–Nsp^2^ HIS1184: 2.752 O4 sp^3^–Osp^2^ HIS1201: 3.156 O2 sp^3^–Osp^2^ HIS1201: 3.002 O2 sp^3^–Osp^2^ ASP1198: 2.678	−74.58/0.82
4W6E in complex with phloretin	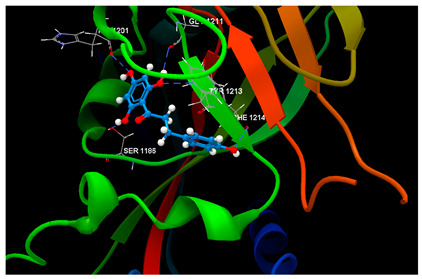	O1 sp^3^–Osp^3^ SER1186: 3.261 O3 sp^3^–Osp^2^ HIS1201: 3.119 O2 sp^3^–Osp^2^ GLY1211: 3.167 O2 sp^3^–Nsp^2^ TYR1213: 3.049 O4 sp^3^–Osp^2^ PHE1214: 3.269	−67.28/0.52
4W6E in complex with 4-methylchalcone		no bonds	−67.80/0.12

**Table 5 ijms-26-03492-t005:** Molecular docking results study on COX-2 target (PDB ID:1PXX) [96].

Test Pairs	The Docking Image of the Test Ligand Interacting with the Amino Acid Residues in the Binding Site	The Punctual Hydrogen Bonds: Å	Score/RMSD
1PXX in complex with diclofenac	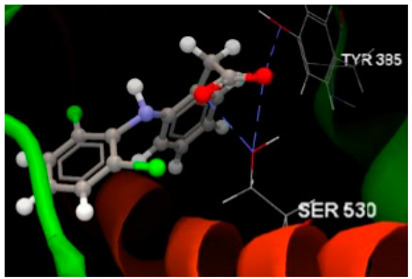	O sp^2^–O sp^3^ SER530 (2.653) O sp^2^–O sp^3^ SER530 (2.905) O sp^2^–O sp^3^ TYR385 (2.729)	−68.76/0.15 [94]
1PXX in complex with phloridzin	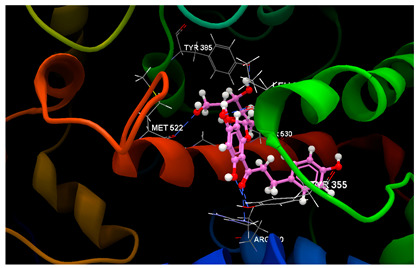	O7 sp^2^–O sp^3^ TYR355 (3.175) O7 sp^2^–N sp^2^ ARG120 (3.155) O9 sp^3^–O sp^3^ TYR355 (2.577) O8 sp^3^–O sp^2^ LEU352 (2.715) O4 sp^3^–O sp^3^ SER530 (3.124) O2 sp^3^–O sp^2^ GLY526 (2.460) O5 sp^3^–O sp^2^ MET522 (2.984)	−52.56/0.34
1PXX in complex with phloretin	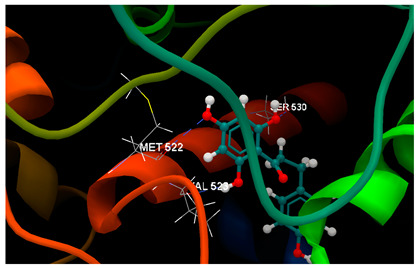	O1 sp^3^–O sp^2^ VAL523 (3.126) O2 sp^3^–O sp^3^ SER530 (2.934) O3 sp^3^–O sp^2^ MET522 (3.244)	−57.24/0.07
1PXX in complex with 4-methylchalcone		no bonds	−58.38/0.09

## Data Availability

The original contributions presented in the study are included in the article/Appendix A; further inquiries can be directed to the corresponding author. The raw data from computations supporting the conclusions of this article will be made available by the authors upon request.

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
