# Peer review of "In Silico and In Vitro Analyses of Strawberry-Derived Extracts in Relation to Key Compounds’ Metabolic and Anti-Tumor Effects"

_ijms, 2025, doi:10.3390/ijms26083492_

Round 1
Reviewer 1 Report
Comments and Suggestions for Authors
- It is recommended to change the semicolon in the title to a comma or reorganize the sentence to improve readability and academic standardization.
- Unify the color scheme of the charts and graphs (for example, avoid high-contrast fluorescent colors) to ensure consistency in the visual style.
- There is no consistent use of line spacing or indentation between paragraphs in the main text, which needs to be adjusted to meet the journal's layout requirements.
- The names of some compounds need to be unified (for example, is "phlordizin" a spelling mistake of "phloridzin"?). It is recommended to check the professional terms and chemical nomenclature rules.
- It is necessary to supplement the existing research evidence on the synergistic effect between strawberry extract and chalcone in the introduction.
- It is recommended to adjust the paragraph order of the article, swapping the "Materials and Methods" section with the "Discussion" section.
Author Response
Thank you very much for your careful review of this article. The paragraphs marked in blue are new data and the paragraphs marked in green are revised data. We tried to answer to all the requests and we are confident that the revised article is significantly improved.
1.It is recommended to change the semicolon in the title to a comma or reorganize the sentence to improve readability and academic standardization. Revised: lines 2-3;
2.Unify the color scheme of the charts and graphs (for example, avoid high-contrast fluorescent colors) to ensure consistency in the visual style. Revised: Figures 5,6,7;
3.There is no consistent use of line spacing or indentation between paragraphs in the main text, which needs to be adjusted to meet the journal's layout requirements. Revised: The article has been reedit using the IJMS template;
4.The names of some compounds need to be unified (for example, is "phlordizin" a spelling mistake of "phloridzin"?). It is recommended to check the professional terms and chemical nomenclature rules. Answer: The article has been fully revised;
5.It is necessary to supplement the existing research evidence on the synergistic effect between strawberry extract and chalcone in the introduction. Revised: We found data regarding the synergism between chalcones and other compounds or therapies (lines 119-132);
6.It is recommended to adjust the paragraph order of the article, swapping the "Materials and Methods" section with the "Discussion" section. Revised: We adjusted the paragraphs’ content and order so as to respond to the observations of all reviewers.
Reviewer 2 Report
Comments and Suggestions for Authors
The present manuscript presents a well-structured study investigating the biological activity of chalcone derivatives and strawberry extracts using in silico and in vitro approaches. The research is of interest to understand the interactions of bioactive compounds with metabolic and cellular pathways. While the study introduces novel concepts, it is imperative to undertake revisions to enhance clarity, methodological accuracy, and scientific impact. The subsequent revisions are aimed at improving the presentation of the results and strengthening the discussion with additional references, thus ensuring that the manuscript is more rigorous and easily understandable to readers.
ABSTRACT
The abstract is well structured and clearly presents the study; however, it should be modified to highlight the relevance of the results in terms of applicability. Instead of concluding with a general summary, an additional sentence should be included to emphasize the practical implications of the findings.
INTRODUCTION
The introduction is well-structured and provides relevant information about chalcones and their derivatives. However, it is necessary to reinforce the justification for the study to clearly present both the initial hypothesis and the study’s objectives.
- Add a paragraph discussing the significance of chalcones in cancer treatments or metabolic disorders.
- Include previous studies that support using in silico models to predict the absorption and metabolism of these compounds.
- Incorporate up-to-date references to strengthen the information provided.
MATERIALS AND METHODS
- Specify the experimental conditions, including incubation times.
- Indicate the number of biological replicates and the techniques used in the cell-based assays.
- The methodology for the in silico studies lacks specificity, particularly regarding the parameters used in molecular modeling software. Clearly define the target protein selection criteria and docking parameters for the in silico studies.
- Provide more details on the cytotoxicity assays in Caco-2 cells, particularly regarding the controls used.
RESULTS AND DISCUSSION
- Improve the description of the results from the cellular assays to emphasize the key findings.
- Expand the comparison of the results obtained in this study with those from previous research, and include references that support the data interpretation.
- Discuss how docking score values may correlate with the effects observed in Caco-2 cells.
- Compare the findings with previous studies on chalcone cytotoxicity in intestinal cell lines.
- More explicitly discuss how the results obtained could impact the development of pharmaceutical or nutraceutical products.
CONCLUSIONS
- The conclusion is too general and does not sufficiently emphasize the potential applications of the study.
- The conclusions should be more specific and focused on the study’s key findings.
- Include potential future research directions that align with this line of study.
- Briefly mention the study's limitations and future research steps.
FIGURES AND TABLES
- Improve the quality of the figures, as some do not appear to meet the minimum recommended resolution.
- Enhance the clarity of the figure legends to facilitate comprehension.

Author Response
Thank you very much for your careful review of this article. The paragraphs marked in blue are new data and the paragraphs marked in green are revised data. We tried to answer to all the requests and we are confident that the revised article is significantly improved.
1.ABSTRACT - The abstract is well structured and clearly presents the study; however, it should be modified to highlight the relevance of the results in terms of applicability. Instead of concluding with a general summary, an additional sentence should be included to emphasize the practical implications of the findings. Revised: The abstract has been rewritten taking these observations into account (lines 18-34);
2. INTRODUCTION -The introduction is well-structured and provides relevant information about chalcones and their derivatives. However, it is necessary to reinforce the justification for the study to clearly present both the initial hypothesis and the study’s objectives. 2.1. Add a paragraph discussing the significance of chalcones in cancer treatments or metabolic disorders. Revised: see lines 89-118; 2.2. Include previous studies that support using in silico models to predict the absorption and metabolism of these compounds. The revision has been made to underline the utility of the in silico studies for practical and scientific purposes (lines 133-149 and 173-178). The SwissADME platform is the unique source to obtain these complete data related to the absorption and the metabolism of the small active compounds. 2.3. Incorporate up-to-date references to strengthen the information provided. Revised: The article contains 25 new references;
3. MATERIALS AND METHODS 3.1. Specify the experimental conditions, including incubation times. Revised: see lines 843-844; 3.2. Indicate the number of biological replicates and the techniques used in the cell-based assays. Revised: see lines 842-851 and 820-822; 3.3. The methodology for the in silico studies lacks specificity, particularly regarding the parameters used in molecular modeling software. Clearly define the target protein selection criteria and docking parameters for the in silico studies. The lines 636-716 in “Discussions” section describe in detail why these 3 molecular targets are relevant for studies and for tumoringenensis and intestinal malignancy in humans. The lines 796-801 and 810-818 in “Materials and methods” section describe the molecular modeling software used, the protein disease criterion selection and the main steps necessary for molecular docking analysis. The extensive description of the CLC analysis on each one molecular target can be found in the previous authors studies (References 52, 104,105). 3.4. Provide more details on the cytotoxicity assays in Caco-2 cells, particularly regarding the controls used. Revised: see lines 837-842 and 389-393;
4. RESULTS AND DISCUSSION 4.1.Improve the description of the results from the cellular assays to emphasize the key findings. Revised: The in vitro study has been rebuild for better understanding (lines 377-513); 4.2. Expand the comparison of the results obtained in this study with those from previous research, and include references that support the data interpretation. Revised: see lines 514-544; 4.3. Discuss how docking score values may correlate with the effects observed in Caco-2 cells. Revised: see lines 727-737; 4.4. Compare the findings with previous studies on chalcone cytotoxicity in intestinal cell lines. Revised: see lines 888-897; 4.5. More explicitly discuss how the results obtained could impact the development of pharmaceutical or nutraceutical products. Answer: similar to the authors of the in vitro study on phloretin lines 888-897, we also made some clarifications regarding the inclusion of the data from in silico and in vitro researches in current practice (lines 902-908).
5. CONCLUSIONS 5.1. The conclusion is too general and does not sufficiently emphasize the potential applications of the study. The conclusions should be more specific and focused on the study’s key findings. Include potential future research directions that align with this line of study. Briefly mention the study's limitations and future research steps. The conclusion section has been fully REVISED.
6. FIGURES AND TABLES 6.1. Improve the quality of the figures, as some do not appear to meet the minimum recommended resolution. Enhance the clarity of the figure legends to facilitate comprehension. Figures and Tables were revised
Reviewer 3 Report
Comments and Suggestions for Authors
The manuscript presents a compelling in silico analysis comparing the pharmacokinetic properties, solubility profiles, medicinal chemistry attributes, and lead-likeness parameters of three major naturally occurring chalcone derivatives (phloridzin, phloretin, and 4-methylchalcone) alongside three small compounds identified in a 40% ethanolic strawberry fruit extract. However, several key refinements could enhance the clarity and impact of the study.
In the Introduction, the statement regarding the study’s primary aim should be revised to accurately reflect that the chalcone derivatives were used as reference compounds rather than being extracted from plant material. This clarification is necessary to prevent any misinterpretation of the study's methodology. Additionally, the manuscript title should be reformulated to eliminate ambiguity and ensure that readers clearly understand the study's focus.
To enhance readability and logical flow, I recommend reorganizing the Results section. Specifically, Table 6 should be positioned before Figure 1 to establish a more structured narrative. The GC-MS results would be more effectively placed in the Results and Discussion section, where their significance can be contextualized. Furthermore, a more intuitive sequence should be followed, starting with Figure 2, then Table 2, followed by Figure 1 and Table 1, to streamline the presentation of findings. Moreover, the manuscript lacks explicit details on the preparation of samples for pharmacological study, specifically 40% ethanolic extract from strawberries, alone and in combination with phloridzin, phloretin, or 4-methylchalcone extraction of the three chalcone derivatives from strawberries, which should be addressed to avoid discrepancies in the methodology.
The Conclusion should be expanded to provide a more comprehensive synthesis of the findings. Additionally, it should incorporate the limitations of the study—such as the constraints of in silico predictions compared to in vitro or in vivo validation—and suggest future research directions, particularly regarding experimental validation of the predicted properties. These refinements will significantly enhance the clarity, accuracy, and overall impact of the manuscript.
Author Response
Thank you very much for your careful review of this article. The paragraphs marked in blue are new data and the paragraphs marked in green are revised data. e tried to answer to all the requests and we are confident that the revised article is significantly improved.
1.In the Introduction, the statement regarding the study’s primary aim should be revised to accurately reflect that the chalcone derivatives were used as reference compounds rather than being extracted from plant material. This clarification is necessary to prevent any misinterpretation of the study's methodology. The Introduction section has been REVISED in both, general aspects (lines 89-153) and the study's aims (lines 154-178);
2.To enhance readability and logical flow, I recommend reorganizing the Results section. Specifically, Table 6 should be positioned before Figure 1 to establish a more structured narrative. The GC-MS results would be more effectively placed in the Results and Discussion section, where their significance can be contextualized. Furthermore, a more intuitive sequence should be followed, starting with Figure 2, then Table 2, followed by Figure 1 and Table 1, to streamline the presentation of findings. REVISED (The article has been fully revised taking into account these recommendations);
3.Moreover, the manuscript lacks explicit details on the preparation of samples for pharmacological study, specifically 40% ethanolic extract from strawberries, alone and in combination with phloridzin, phloretin, or 4-methylchalcone extraction of the three chalcone derivatives from strawberries, which should be addressed to avoid discrepancies in the methodology. REVISED: see lines 820-850;
4.The Conclusion should be expanded to provide a more comprehensive synthesis of the findings. REVISED (The conclusions section has been completely revised - see lines 853-908);
5.Additionally, it should incorporate the limitations of the study—such as the constraints of in silico predictions compared to in vitro or in vivo validation (REVISED: lines 727-737) and suggest future research directions (REVISED: lines 898-908), particularly regarding experimental validation of the predicted properties (SwissADME tool provides very precise data regarding the experimental validation of each one parameter studied). These refinements will significantly enhance the clarity, accuracy, and overall impact of the manuscript.
Round 2
Reviewer 1 Report
Comments and Suggestions for Authors
- The discussion section requires further in - depth exploration. It is recommended to conduct an in - depth analysis of the clinical significance and rigorously verify the mechanism.
- It is recommended to add a dedicated "Limitations" subsection in the "Discussion" or "Conclusion" section to clearly state the deficiencies of the current study and propose directions for future improvement.
- Conduct a systematic optimization of the language throughout the text, standardize the terminology and abbreviations, simplify long sentences, and enhance the use of logical conjunctions.
Author Response
1.The discussion section requires further in - depth exploration. It is recommended to conduct an in - depth analysis of the clinical significance and rigorously verify the mechanism. REVISED: lines 729-781 plus the references from 164 to 175;
2.It is recommended to add a dedicated "Limitations" subsection in the "Discussion" or "Conclusion" section to clearly state the deficiencies of the current study and propose directions for future improvement. REVISED: lines 954-967. Also, the in vitro-in silico data corroboration has been moved in “Conclusion” section: lines 926-935.
3.Conduct a systematic optimization of the language throughout the text, standardize the terminology and abbreviations, simplify long sentences, and enhance the use of logical conjunctions. REVISED: ALL PAPER.

Reviewer 3 Report
Comments and Suggestions for Authors
Agree with revised manuscript.
Author Response
At the reviewers recommendation, 2 new sections (lines 729-781 and 954-967) plus 11 new references (from 164 to 175) have been added.
We hope that now the article can be fully understood.
Thank you very much!
